# CLimODE: Climate and Weather Forecasting with Physics-informed Neural ODEs

**Yogesh Verma, Markus Heinonen**
Department of Computer Science
Aalto University, Finland
{yogesh.verma,markus.o.heinonen}@aalto.fi

**Vikas Garg**
YaiYai Ltd and Aalto University
vgarg@csail.mit.edu

## ABSTRACT

Climate and weather prediction traditionally relies on complex numerical simulations of atmospheric physics. Deep learning approaches, such as transformers, have recently challenged the simulation paradigm with complex network forecasts. However, they often act as data-driven black-box models that neglect the underlying physics and lack uncertainty quantification. We address these limitations with ClimODE, a spatiotemporal continuous-time process that implements a key principle of *advection* from statistical mechanics, namely, weather changes due to a spatial movement of quantities over time. ClimODE models precise weather evolution with value-conserving dynamics, learning global weather transport as a neural flow, which also enables estimating the uncertainty in predictions. Our approach outperforms existing data-driven methods in global and regional forecasting with an order of magnitude smaller parameterization, establishing a new state of the art.

## 1 INTRODUCTION

State-of-the-art climate and weather prediction relies on high-precision numerical simulation of complex atmospheric physics (Phillips, 1956; Satoh, 2004; Lynch, 2008). While accurate to medium timescales, they are computationally intensive and largely proprietary (NOAA, 2023; ECMWF, 2023).

There is a long history of 'free-form' neural networks challenging the mechanistic simulation paradigm (Kuligowski & Barros, 1998; Baboo & Shereef, 2010), and recently deep learning has demonstrated significant successes (Nguyen et al., 2023). These methods range from one-shot GANs (Ravuri et al., 2021) to autoregressive transformers (Pathak et al., 2022; Nguyen et al., 2023; Bi et al., 2023) and multi-scale GNNs (Lam et al., 2022). Zhang et al. (2023) combines autoregression with physics-inspired transport flow.

In statistical mechanics, weather can be described as a *flux*, a spatial movement of quantities over time, governed by the partial differential *continuity equation* (Broomé & Ridenour, 2014)

$$\underbrace{\frac{du}{dt}}_{\text{time evolution } \dot{u}} + \overbrace{\underbrace{\mathbf{v} \cdot \nabla u}^{\text{transport}} + \underbrace{\overbrace{u \nabla \cdot \mathbf{v}}^{\text{compression}}}_{\text{advection}}} = \underbrace{s}_{\text{sources}}, \tag{1}$$

where $u(\mathbf{x}, t)$ is a quantity (e.g. temperature) evolving over space $\mathbf{x} \in \Omega$ and time $t \in \mathbb{R}$ driven by a flow's velocity $\mathbf{v}(\mathbf{x}, t) \in \Omega$ and sources $s(\mathbf{x}, t)$ (see Figure 1). The advection moves and redistributes existing weather 'mass' spatially, while sources add or remove quantities. Crucially, the dynamics need to be continuous-time, and modeling them with autoregressive 'jumps' violates the conservation of mass and incurs approximation errors.

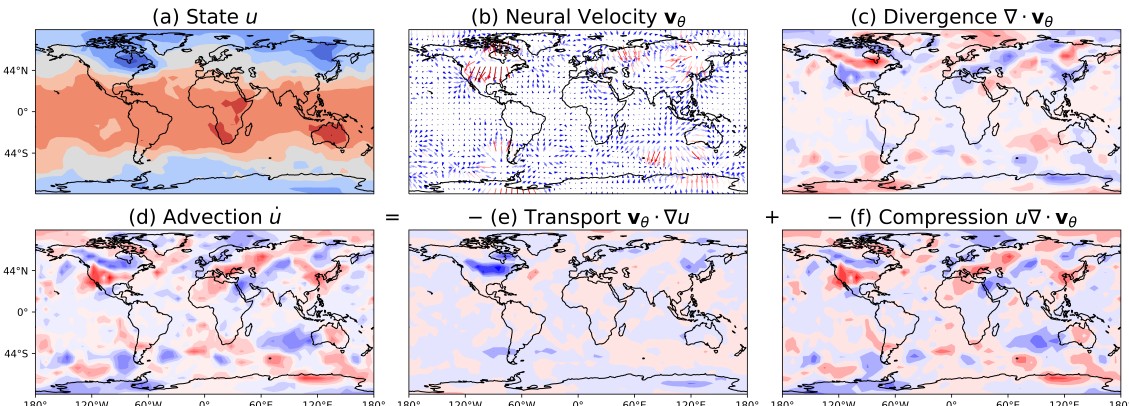

Figure 1: **Weather as a quantity-preserving advection system.** A quantity (eg. temperature) **(a)** is moved by a neural flow velocity **(b)**, whose divergence is the flow's compressibility **(c)**. The flow translates into state change by advection **(d)**, which combine quantity's transport **(e)** and compression **(f)**.

We introduce a climate model that implements a continuous-time, second-order neural continuity equation with simple yet powerful inductive biases that ensure – by definition – value-conserving dynamics with more stable long-horizon forecasts. We show a computationally practical method to solve the continuity equation over entire Earth as a system of neural ODEs. We learn the flow **v** as a neural network with only a few million parameters that uses both global attention and local convolutions. Furthermore, we address source variations via a probabilistic emission model that quantifies prediction uncertainties. Empirical evidence underscores ClimODE's ability to attain state-of-the-art global and regional weather forecasts.

## 1.1 CONTRIBUTIONS

We propose to learn a continuous-time PDE model, grounded on physics, for climate and weather modeling and uncertainty quantification. In particular,

- we propose ClimODE, a continuous-time neural advection PDE climate and weather model, and derive its ODE system tailored to numerical weather prediction.
- we introduce a flow velocity network that integrates local convolutions, long-range attention in the ambient space, and a Gaussian emission network for predicting uncertainties and source variations.
- empirically, ClimODE achieves state-of-the-art global and regional forecasting performance.
- Our physics-inspired model enables efficient training from scratch on a single GPU and comes with an open-source PyTorch implementation on GitHub.[1]

## 2 RELATED WORKS

**Numerical climate and weather models.** Current models encompass numerical weather prediction (NWP) for short-term weather forecasts and climate models for long-term climate predictions. The cutting-edge approach in climate modeling involves earth system models (ESM) (Hurrell et al., 2013), which integrate simulations of physics of the atmosphere, cryosphere, land, and ocean processes. While successful,

---

[1]https://github.com/Aalto-QuML/ClimODE

Table 1: **Overview of current deep learning methods for weather forecasting.**

| Method | Value-preserving | Explicit Periodicity/Seasonality | Uncertainty | Continuous-time | Parameters (M) | |
|---|---|---|---|---|---|---|
| FourCastNet | ✗ | ✗ | ✗ | ✗ | N/A | Pathak et al. (2022) |
| GraphCast | ✗ | ✗ | ✗ | ✗ | 37 | Lam et al. (2022) |
| Pangu-Weather | ✗ | ✗ | ✗ | ✗ | 256 | Bi et al. (2023) |
| ClimaX | ✗ | ✗ | ✗ | ✗ | 107 | Nguyen et al. (2023) |
| NowcastNet | ✓ | ✗ | ✗ | ✗ | N/A | Zhang et al. (2023) |
| ClimODE | ✓ | ✓ | ✓ | ✓ | 2.8 | this work |

they exhibit sensitivity to initial conditions, structural discrepancies across models (Balaji et al., 2022), regional variability, and high computational demands.

**Deep learning for forecasting.** Deep learning has emerged as a compelling alternative to NWPs, focusing on global forecasting tasks. Rasp et al. (2020) employed pre-training techniques using ResNet (He et al., 2016) for effective medium-range weather prediction, Weyn et al. (2021) harnessed a large ensemble of deep-learning models for sub-seasonal forecasts, whereas Ravuri et al. (2021) used deep generative models of radar for precipitation nowcasting and GraphCast (Lam et al., 2022; Keisler, 2022) utilized a graph neural network-based approach for weather forecasting. Additionally, recent state-of-the-art neural forecasting models of ClimaX (Nguyen et al., 2023), FourCastNet (Pathak et al., 2022), and Pangu-Weather (Bi et al., 2023) are predominantly built upon data-driven backbones such as Vision Transformer (ViT) (Dosovitskiy et al., 2021), UNet (Ronneberger et al., 2015), and autoencoders. However, these models overlook the fundamental physical dynamics and do not offer uncertainty estimates for their predictions.

**Neural ODEs.** Neural ODEs propose learning the time derivatives as neural networks (Chen et al., 2018; Massaroli et al., 2020), with multiple extensions to adding physics-based constraints (Greydanus et al., 2019; Cranmer et al., 2020; Brandstetter et al., 2023; Choi et al., 2023). The physics-inspired networks (PINNs) embed mechanistic understanding in neural ODEs (Raissi et al., 2019; Cuomo et al., 2022), while multiple lines of works attempt to uncover interpretable differential forms (Brunton et al., 2016; Fronk & Petzold, 2023). Neural PDEs warrant solving the system through spatial discretization (Poli et al., 2019; Iakovlev et al., 2021) or functional representation (Li et al., 2021). Machine learning has also been used to enhance fluid dynamics models (Li et al., 2021; Lu et al., 2021; Kochkov et al., 2021). The above methods are predominantly applied to only small, non-climate systems.

## 3 NEURAL TRANSPORT MODEL

**Notation.** Throughout the paper $\nabla = \nabla_{\mathbf{x}}$ denotes spatial gradients, $\dot{u} = \frac{du}{dt}$ time derivatives, $\cdot$ inner product, and $\nabla \cdot \mathbf{v} = \text{tr}(\nabla \mathbf{v})$ divergence. We color equations purely for cosmetic clarity.

### 3.1 ADVECTION EQUATION

We model weather as a spatiotemporal process $\mathbf{u}(\mathbf{x}, t) = (u_1(\mathbf{x}, t), \dots, u_K(\mathbf{x}, t)) \in \mathbb{R}^K$ of $K$ quantities $u_k(\mathbf{x}, t) \in \mathbb{R}$ over continuous time $t \in \mathbb{R}$ and latitude-longitude locations $\mathbf{x} = (h, w) \in \Omega = [-90°, 90°] \times [-180°, 180°] \subset \mathbb{R}^2$. We assume the process follows an advection partial differential equation

$$\dot{u}_k(\mathbf{x}, t) = -\underbrace{\mathbf{v}_k(\mathbf{x}, t) \cdot \nabla u_k(\mathbf{x}, t)}_{\text{transport}} - \underbrace{u_k(\mathbf{x}, t) \nabla \cdot \mathbf{v}_k(\mathbf{x}, t)}_{\text{compression}}, \qquad (2)$$

where quantity change $\dot{u}_k(\mathbf{x}, t)$ is caused by the flow, whose velocity $\mathbf{v}_k(\mathbf{x}, t) \in \Omega$ transports and concentrates air mass (see Figure 2). The equation (2) describes a *closed* system, where value $u_k$ is moved around

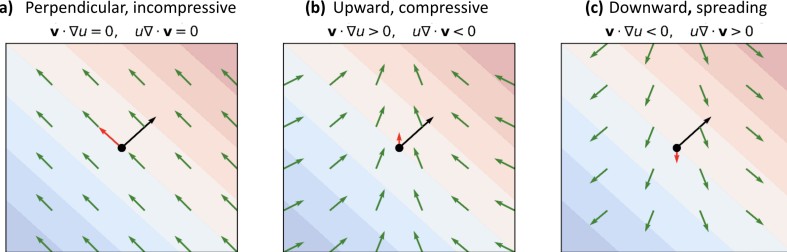

Figure 2: **Conceptual illustration of continuity equation on pointwise temperature change** $\dot{u}(\mathbf{x}_0, t) = -\mathbf{v} \cdot \nabla u - u \nabla \cdot \mathbf{v}$. **(a)** A perpendicular flow (green) to the gradient (blue to red) moves in equally hot air causing no change at $\mathbf{x}_0$. **(b)** Cool air moves upwards, decreasing pointwise temperature, while air concentration at $\mathbf{x}_0$ accumulates additional temperature. **(c)** Hot air moves downwards increasing temperature at $\mathbf{x}_0$, while air dispersal decreases it.

but never lost or added. While a realistic assumption on average, we will introduce an emission source model in Section 3.7. The closed system assumption forces the simulated trajectories $u_k(\mathbf{x}, t)$ to *value-preserving* manifold

$$\int u_k(\mathbf{x}, t) d\mathbf{x} = \text{const}, \qquad \forall t, k. \tag{3}$$

This is a strong inductive bias that prevents long-horizon forecast collapses (see Appendix H for details.)

## 3.2 FLOW VELOCITY

Next, we need a way to model the flow velocity $\mathbf{v}(\mathbf{x}, t)$ (See Figure 1b). Earlier works have remarked that second-order bias improves the performance of neural ODEs significantly (Yildiz et al., 2019; Gruver et al., 2022). Similarly, we propose a second-order flow by parameterizing the change of velocity with a neural network $f_\theta$,

$$\dot{\mathbf{v}}_k(\mathbf{x}, t) = f_\theta\Big(\mathbf{u}(t), \nabla \mathbf{u}(t), \mathbf{v}(t), \psi\Big), \tag{4}$$

as a function of the current state $\mathbf{u}(t) = \{\mathbf{u}(\mathbf{x}, t) : \mathbf{x} \in \Omega\} \in \mathbb{R}^{K \times H \times W}$, its gradients $\nabla \mathbf{u}(t) \in \mathbb{R}^{2K \times H \times W}$, the current velocity $\mathbf{v}(t) = \{\mathbf{v}(\mathbf{x}, t) : \mathbf{x} \in \Omega\} \in \mathbb{R}^{2K \times H \times W}$, and spatiotemporal embeddings $\psi \in \mathbb{R}^{C \times H \times W}$. These inputs denote global *frames* (e.g., Figure 1) at time $t$ discretized to a resolution $(H, W)$ with a total of $5K$ quantity channels and $C$ embedding channels.

## 3.3 2ND-ORDER PDE AS A SYSTEM OF FIRST-ORDER ODES

We utilize the method of lines (MOL), discretizing the PDE into a grid of location-specific ODEs (Schiesser, 2012; Iakovlev et al., 2021). Additionally, a second-order differential equation can be transformed into a pair of first-order differential equations (Kreyszig, 2020; Yildiz et al., 2019). Combining these techniques yields a system of first-order ODEs $(u_{ki}(t), v_{ki}(t))$ of quantities $k$ at locations $\mathbf{x}_i$:

$$\begin{bmatrix} \mathbf{u}(t) \\ \mathbf{v}(t) \end{bmatrix} = \begin{bmatrix} \mathbf{u}(t_0) \\ \mathbf{v}(t_0) \end{bmatrix} + \int_{t_0}^t \begin{bmatrix} \dot{\mathbf{u}}(\tau) \\ \dot{\mathbf{v}}(\tau) \end{bmatrix} d\tau = \begin{bmatrix} \{u_k(t_0)\}_k \\ \{\mathbf{v}_k(t_0)\}_k \end{bmatrix} + \int_{t_0}^t \begin{bmatrix} \left\{ -\nabla \cdot (u_k(\tau)\mathbf{v}_k(\tau)) \right\}_k \\ \left\{ f_\theta\big(\mathbf{u}(\tau), \nabla \mathbf{u}(\tau), \mathbf{v}(\tau), \psi\big)_k \right\}_k \end{bmatrix} d\tau, \tag{5}$$

where $\tau \in \mathbb{R}$ is an integration time, and where we apply equations (2) and (4). Backpropagation of ODEs is compatible with standard autodiff, while also admitting tractable adjoint form (LeCun et al., 1988; Chen et al., 2018; Metz et al., 2021). The forward solution $\mathbf{u}(t)$ can be accurately approximated with numerical solvers such as Runge-Kutta (Runge, 1895) with low computational cost.

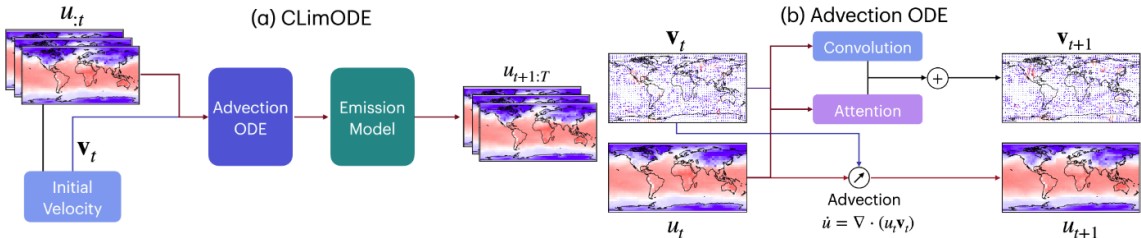

Figure 3: Whole prediction pipeline for ClimODE.

### 3.4 Modeling local and global effects

PDEs link acceleration $\dot{\mathbf{v}}(\mathbf{x}, t)$ solely to the current state and its gradient at the same location $\mathbf{x}$ and time $t$, ruling out long-range connections. However, long-range interactions naturally arise as information propagates over time across substantial distances. For example, Atlantic weather conditions influence future weather patterns in Europe and Africa, complicating the covariance relationships between these regions. Therefore, we propose a hybrid network to account for both local transport and global effects,

$$f_\theta\Big(\mathbf{u}(t), \nabla\mathbf{u}(t), \mathbf{v}(t), \psi\Big) = \underbrace{f_{\mathrm{conv}}\Big(\mathbf{u}(t), \nabla\mathbf{u}(t), \mathbf{v}(t), \psi\Big)}_{\text{convolution network}} + \gamma\,\underbrace{f_{\mathrm{att}}\Big(\mathbf{u}(t), \nabla\mathbf{u}(t), \mathbf{v}(t), \psi\Big)}_{\text{attention network}}. \qquad (6)$$

**Local Convolutions** To capture local effects, we employ a *local* convolution network, denoted as $f_{\mathrm{conv}}$. This network is parameterized using ResNets with 3x3 convolution layers, enabling it to aggregate weather information up to a distance of $L$ 'pixels' away from the location $\mathbf{x}$, where $L$ corresponds to the network's depth. Additional parameterization details can be found in Appendix C.

**Attention Convolutional Network** We include an attention convolutional network $f_{\mathrm{att}}$ which captures *global* information by considering states across the entire Earth, enabling long-distance connections. This attention network is structured around KQV dot product, with Key, Query, and Value parameterized with CNNs. Further elaboration is provided in Appendix C.2 and $\gamma$ is a learnable hyper-parameter.

### 3.5 Spatiotemporal embedding

**Day and Season** We encode daily and seasonal periodicity of time $t$ with trigonometric time embeddings

$$\psi(t) = \left\{ \sin 2\pi t, \cos 2\pi t, \sin\frac{2\pi t}{365}, \cos\frac{2\pi t}{365} \right\}. \qquad (7)$$

**Location** We encode latitude $h$ and longitude $w$ with trigonometric and spherical-position encodings

$$\psi(\mathbf{x}) = \big[\{\sin, \cos\} \times \{h, w\}, \sin(h)\cos(w), \sin(h)\sin(w)\big]. \qquad (8)$$

**Joint time-location embedding** We create a joint location-time embedding by combining position and time encodings $(\psi(t) \times \psi(\mathbf{x}))$, capturing the cyclical patterns of day and season across different locations on the map. Additionally, we incorporate constant spatial and time features, with $\psi(h)$ and $\psi(w)$ representing 2D latitude and longitude maps, and lsm and oro denoting static variables in the data,

$$\psi(\mathbf{x}, t) = \big[\psi(t), \psi(\mathbf{x}), \psi(t) \times \psi(\mathbf{x}), \psi(c)\big], \qquad \psi(c) = \big[\psi(h), \psi(w), \mathrm{lsm}, \mathrm{oro}\big]. \qquad (9)$$

These spatiotemporal features are additional input channels to the neural networks (See Appendix B).

### 3.6 INITIAL VELOCITY INFERENCE

The neural transport model necessitates an initial velocity estimate, $\hat{\mathbf{v}}_k(\mathbf{x}, t_0)$, to start the ODE solution (5). In traditional dynamic systems, estimating velocity poses a challenging inverse problem, often requiring encoders in earlier neural ODEs (Chen et al., 2018; Yildiz et al., 2019; Rubanova et al., 2019; De Brouwer et al., 2019). In contrast, the continuity Equation (2) establishes an identity, $\dot{u} + \nabla \cdot (u\mathbf{v}) = 0$, allowing us to solve directly for the missing velocity, $\mathbf{v}$, when observing the state $u$. We optimize the initial velocity for location $\mathbf{x}$, time $t$ and quantity $k$ with penalised least-squares

$$\hat{\mathbf{v}}_k(t) = \underset{\mathbf{v}_k(t)}{\arg\min} \left\{ \left\| \tilde{\dot{u}}_k(t) + \mathbf{v}_k(t) \cdot \tilde{\nabla} u_k(t) + u_k(t) \tilde{\nabla} \cdot \mathbf{v}_k(\mathbf{x}, t) \right\|_2^2 + \alpha \left\| \mathbf{v}_k(t) \right\|_{\mathbf{K}} \right\}, \qquad (10)$$

where $\tilde{\nabla}$ is numerical spatial derivative, and $\tilde{\dot{u}}(t_0)$ is numerical approximation from the past states $u(t < t_0)$. We include a Gaussian prior $\mathcal{N}(\text{vec } \mathbf{v}_k | \mathbf{0}, \mathbf{K})$ with a Gaussian RBF kernel $\mathbf{K}_{ij} = \text{rbf}(\mathbf{x}_i, \mathbf{x}_j)$ that results in spatially smooth initial velocities with smoothing coefficient $\alpha$. See Appendix D.5 for details.

### 3.7 SYSTEM SOURCES AND UNCERTAINTY ESTIMATION

The model described so far has two limitations: (i) the system is deterministic and thus has no uncertainty, and (ii) the system is closed and does not allow value loss or gain (eg. during day-night cycle). We tackle both issues with an emission $g$ outputting a bias $\mu_k(\mathbf{x}, t)$ and variance $\sigma_k^2(\mathbf{x}, t)$ of $u_k(\mathbf{x}, t)$ as a Gaussian,

$$u_k^{\text{obs}}(\mathbf{x}, t) \sim \mathcal{N}\Big( u_k(\mathbf{x}, t) + \mu_k(\mathbf{x}, t), \sigma_k^2(\mathbf{x}, t) \Big), \qquad \mu_k(\mathbf{x}, t), \sigma_k(\mathbf{x}, t) = g_k\big(\mathbf{u}(\mathbf{x}, t), \psi\big). \qquad (11)$$

The variances $\sigma_k^2$ represent the uncertainty of the climate estimate, while the mean $\mu_k$ represents value gain bias. For instance, the $\mu$ can model the fluctuations in temperature during the day-night cycle. This can be regarded as an emission model, accounting for the total aleatoric and epistemic variance.

### 3.8 LOSS

We assume a full-earth dataset $\mathcal{D} = (\mathbf{y}_1, \dots, \mathbf{y}_N)$ of a total of $N$ timepoints of observed frames $\mathbf{y}_i \in \mathbb{R}^{K \times H \times W}$ at times $t_i$. We assume the data is organized into a dense and regular spatial grid $(H, W)$, a common data modality. We minimize the negative log-likelihood of the observations $\mathbf{y}_i$,

$$\mathcal{L}(\theta; \mathcal{D}) = -\frac{1}{NKHW} \sum_{i=1}^{N} \left( \log \mathcal{N}\Big(\mathbf{y}_i | \mathbf{u}(t_i) + \boldsymbol{\mu}(t_i), \text{diag}\,\boldsymbol{\sigma}^2(t_i)\Big) + \log \mathcal{N}_+\big(\boldsymbol{\sigma}(t_i) | \mathbf{0}, \lambda_\sigma^2 I\big) \right), \qquad (12)$$

where we also add a Gaussian prior for the variances with a hypervariance $\lambda_\sigma$ to prevent variance explosion during training. We decay the $\lambda_\sigma^{-1}$ using cosine annealing during training to remove its effects and arrive at a maximum likelihood estimate. Further details are provided in Appendix D.

## 4 EXPERIMENTS

**Tasks** We assess ClimODE's forecasting capabilities by predicting the future state $\mathbf{u}_{t+\Delta t}$ based on the initial state $\mathbf{u}_t$ for lead times ranging from $\Delta t = 6$ to $36$ hours both global and regional weather prediction, and monthly average states for climate forecasting. Our evaluation encompasses global, regional and climate forecasting, as discussed in Sections 4.1, 4.2 and 4.3, focusing on key meteorological variables.

**Data.** We use the preprocessed $5.625°$ resolution and 6 hour increment ERA5 dataset from WeatherBench (Rasp et al., 2020) in all experiments. We consider $K = 5$ quantities from the ERA5 dataset: ground

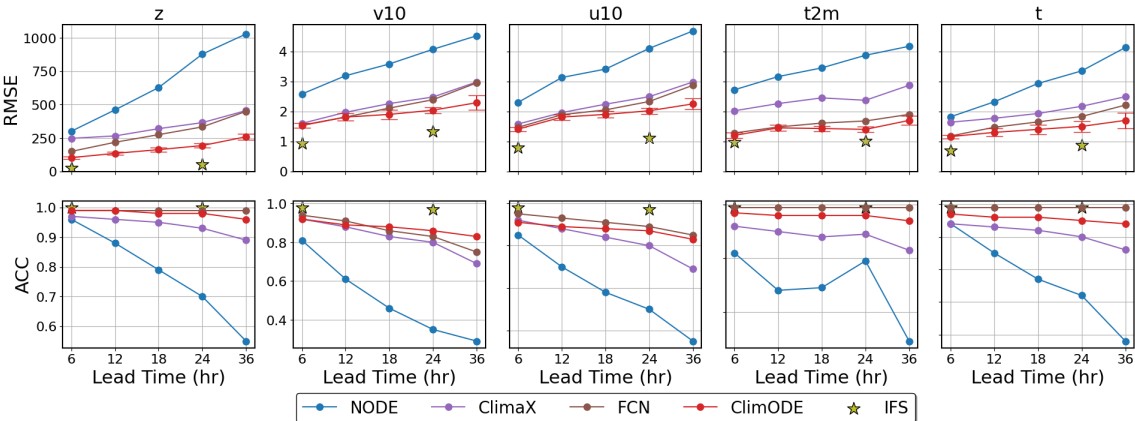

Figure 4: RMSE(↓) and ACC(↑) comparison with baselines. **ClimODE** outperforms competitive neural methods across different metrics and variables. For more details, see Table 6.

temperature (t2m), atmospheric temperature (t), geopotential (z), and ground wind vector (u10, v10) and normalize the variables to $[0, 1]$ via min-max scaling. Notably, both z and t hold standard importance as verification variables in medium-range Numerical Weather Prediction (NWP) models, while t2m and (u10, v10) directly pertain to human activities. We use ten years of training data (2006-15), the validation data is 2016 as validation, and two years 2017-18 as testing data. More details can be found in Appendix B.

**Metrics.** We assess benchmarks using latitude-weighted RMSE and Anomaly Correlation Coefficient (ACC) following the de-normalization of predictions.

$$\text{RMSE} = \frac{1}{N}\sum_{t}^{N}\sqrt{\frac{1}{HW}\sum_{h}^{H}\sum_{w}^{W}\alpha(h)(y_{thw}-u_{thw})^2}, \ \text{ACC} = \frac{\sum_{t,h,w}\alpha(h)\tilde{y}_{thw}\tilde{u}_{thw}}{\sqrt{\sum_{t,h,w}\alpha(h)\tilde{y}_{thw}^2\sum_{t,h,w}\alpha(h)\tilde{u}_{thw}^2}} \quad (13)$$

where $\alpha(h) = \cos(h)/\frac{1}{H}\sum_{h'}^{H}\cos(h')$ is the latitude weight and $\tilde{y} = y - C$ and $\tilde{u} = u - C$ are averaged against empirical mean $C = \frac{1}{N}\sum_t y_{thw}$. More detail in Appendix C.3.

**Competing methods.** Our method is benchmarked against exclusively open-source counterparts. We compare primarily against **ClimaX** (Nguyen et al., 2023), a state-of-the-art Transformer method trained on same dataset, **FourCastNet (FCN)** (Pathak et al., 2022), a large-scale model based on adaptive fourier neural operators and against a **Neural ODE**. We were unable to compare with PanguWeather (Bi et al., 2023) and GraphCast (Lam et al., 2022) due to unavailability of their code during the review period. We ensure fairness by retraining all methods from scratch using identical data and variables without pre-training.

**Gold-standard benchmark.** We also compare to the Integrated Forecasting System **IFS** (ECMWF, 2023), one of the most advanced global physics simulation model, often known as simply the 'European model'. Despite its high computational demands, various machine learning techniques have shown superior performance over the IFS, as evidenced (Ben Bouallegue et al., 2024), particularly when leveraging a multitude of variables and exploiting correlations among them, our study focuses solely on a limited subset of these variables, with IFS serving as the gold standard. More details can be found in Appendix D.

Table 2: RMSE($\downarrow$) comparison with baselines for regional forecasting. **ClimODE** outperforms other competing methods in `t2m,t,z` and achieves competitive performance on `u10,v10` across all regions.

| Value | Hours | North-America | | | South-America | | | Australia | | |
|---|---|---|---|---|---|---|---|---|---|---|
| | | NODE | ClimaX | ClimODE | NODE | ClimaX | ClimODE | NODE | ClimaX | ClimODE |
| z | 6 | 232.8 | 273.4 | **134.5** $\pm$ 10.6 | 225.60 | 205.40 | **107.7** $\pm$ 20.2 | 251.4 | 190.2 | **103.8** $\pm$ 14.6 |
| | 12 | 469.2 | 329.5 | **225.0** $\pm$ 17.3 | 365.6 | 220.15 | **169.4** $\pm$ 29.6 | 344.8 | 184.7 | **170.7** $\pm$ 21.0 |
| | 18 | 667.2 | 543.0 | **307.7** $\pm$ 25.4 | 551.9 | 269.24 | **237.8** $\pm$ 32.2 | 539.9 | 222.2 | **211.1** $\pm$ 31.6 |
| | 24 | 893.7 | 494.8 | **390.1** $\pm$ 32.3 | 660.3 | 301.81 | **292.0** $\pm$ 38.9 | 632.7 | 324.9 | **308.2** $\pm$ 30.6 |
| t | 6 | 1.96 | 1.62 | **1.28** $\pm$ 0.06 | 1.58 | 1.38 | **0.97** $\pm$ 0.13 | 1.37 | 1.19 | **1.05** $\pm$ 0.12 |
| | 12 | 3.34 | 1.86 | **1.81** $\pm$ 0.13 | 2.18 | 1.62 | **1.25** $\pm$ 0.18 | 2.18 | 1.30 | **1.20** $\pm$ 0.16 |
| | 18 | 4.21 | 2.75 | **2.03** $\pm$ 0.16 | 2.74 | 1.79 | **1.43** $\pm$ 0.20 | 2.68 | 1.39 | **1.33** $\pm$ 0.21 |
| | 24 | 5.39 | 2.27 | **2.23** $\pm$ 0.18 | 3.41 | 1.97 | **1.65** $\pm$ 0.26 | 3.32 | 1.92 | **1.63** $\pm$ 0.24 |
| t2m | 6 | 2.65 | 1.75 | **1.61** $\pm$ 0.2 | 2.12 | 1.85 | **1.33** $\pm$ 0.26 | 1.88 | 1.57 | **0.80** $\pm$ 0.13 |
| | 12 | 3.43 | **1.87** | 2.13 $\pm$ 0.37 | 2.42 | 2.08 | **1.04** $\pm$ 0.17 | 2.02 | 1.57 | **1.10** $\pm$ 0.22 |
| | 18 | 3.53 | 2.27 | **1.96** $\pm$ 0.33 | 2.60 | 2.15 | **0.98** $\pm$ 0.17 | 3.51 | 1.72 | **1.23** $\pm$ 0.24 |
| | 24 | 3.39 | **1.93** | 2.15 $\pm$ 0.20 | 2.56 | 2.23 | **1.17** $\pm$ 0.26 | 2.46 | 2.15 | **1.25** $\pm$ 0.25 |
| u10 | 6 | 1.96 | 1.74 | **1.54** $\pm$ 0.19 | 1.94 | 1.27 | **1.25** $\pm$ 0.18 | 1.91 | 1.40 | **1.35** $\pm$ 0.17 |
| | 12 | 2.91 | 2.24 | **2.01** $\pm$ 0.20 | 2.74 | 1.57 | **1.49** $\pm$ 0.23 | 2.86 | **1.77** | 1.78 $\pm$ 0.21 |
| | 18 | 3.40 | 3.24 | **2.17** $\pm$ 0.34 | 3.24 | 1.83 | **1.81** $\pm$ 0.29 | 3.44 | 2.03 | **1.96** $\pm$ 0.25 |
| | 24 | 3.96 | 3.14 | **2.34** $\pm$ 0.32 | 3.77 | 2.04 | **2.08** $\pm$ 0.35 | 3.91 | 2.64 | **2.33** $\pm$ 0.33 |
| v10 | 6 | 2.16 | 1.83 | **1.67** $\pm$ 0.23 | 2.29 | 1.31 | **1.30** $\pm$ 0.21 | 2.38 | 1.47 | **1.44** $\pm$ 0.20 |
| | 12 | 3.20 | 2.43 | **2.03** $\pm$ 0.31 | 3.42 | **1.64** | 1.71 $\pm$ 0.28 | 3.60 | **1.79** | 1.87 $\pm$ 0.26 |
| | 18 | 3.96 | 3.52 | **2.31** $\pm$ 0.37 | 4.16 | **1.90** | 2.07 $\pm$ 0.31 | 4.31 | 2.33 | **2.23** $\pm$ 0.23 |
| | 24 | 4.57 | 3.39 | **2.50** $\pm$ 0.41 | 4.76 | **2.14** | 2.43 $\pm$ 0.34 | 4.88 | 2.58 | **2.53** $\pm$ 0.32 |

## 4.1 GLOBAL WEATHER FORECASTING

We assess ClimODE's performance in global forecasting, encompassing the prediction of crucial meteorological variables described above. Figure 4 and Table 6 demonstrate ClimODE's superior performance across all metrics and variables over other neural baselines, while falling short against the gold-standard IFS, as expected. Fig. 10 reports CRPS (Continuous Ranked Probability Score) over the predictions. These findings indicate the effectiveness of incorporating an underlying physical framework for weather modeling.

## 4.2 REGIONAL WEATHER FORECASTING

We assess ClimODE's performance in regional forecasting, constrained to the bounding boxes of North America, South America, and Australia, representing diverse Earth regions. Table 2 reveals noteworthy outcomes. ClimODE has superior predictive capabilities in forecasting ground temperature (`t2m`), atmospheric temperature (`t`), and geopotential (`z`). It also maintains competitive performance in modeling ground wind vectors (`u10` and `v10`) across these varied regions. This underscores ClimODE's proficiency in effectively modeling regional weather dynamics.

## 4.3 CLIMATE FORECASTING: MONTHLY AVERAGE FORECASTING

To demonstrate the versatility of our method, we assess its performance in climate forecasting. Climate forecasting entails predicting the average weather conditions over a defined period. In our evaluation, we focus on monthly forecasts, predicting the average values of key meteorological variables over one-month durations. We maintained consistency by utiliz the same ERA5 dataset and variables employed in previous experiments, and trained the model with same hyperparameters. Our comparative analysis with FourCastNet on latitude-weighted RMSE and ACC is illustrated in Figure 10. Notably, ClimODE demonstrates significantly improved monthly predictions as compared to FourCastNet showing efficacy in climate forecasting.

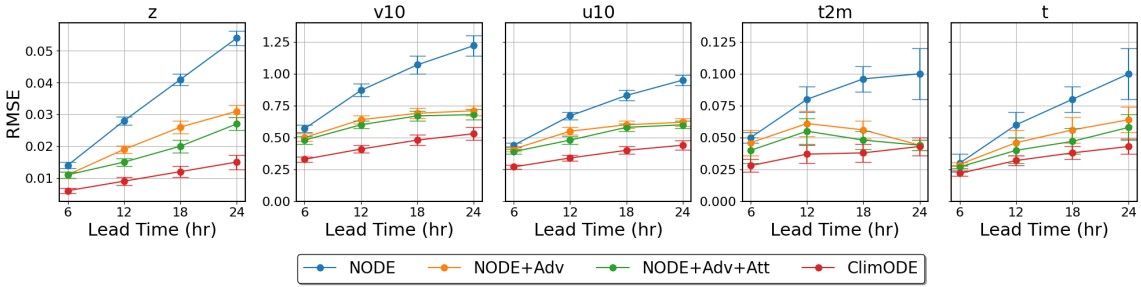

Figure 5: **Effect of Individual Components**: The importance of individual model components. An ablation showing how iteratively enhancing the vanilla neural ODE (blue) with advection form (orange), global attention (green), and emission (red), improves performance of ClimODE. The advection component brings about the most accuracy improvements, while attention turns out to be least important.

## 5  ABLATION STUDIES

**Effect of emission model**  Figure 6 shows model predictions $u(\mathbf{x}, t)$ of ground temperature (t2m) for a specific location while also including emission bias $\mu(\mathbf{x}, t)$ and variance $\sigma^2(\mathbf{x}, t)$. Remarkably, the model captures diurnal variations and effectively estimates variance. Figure 8 highlights bias and variance on a global scale. Positive bias is evident around the Pacific ocean, corresponding to daytime, while negative bias prevails around Europe and Africa, signifying nighttime. The uncertainties indicate confident ocean estimation, with northern regions being challenging.

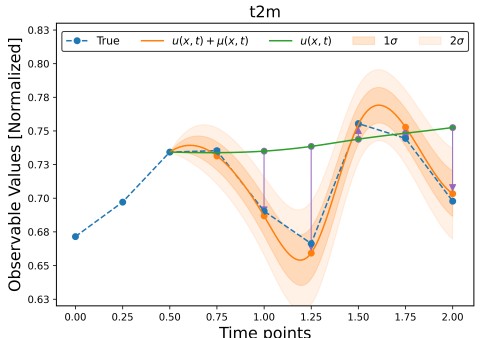

Figure 6: **Effect of bias**: t2m observed and predicted values showcasing the effect of bias.

**Effect of individual components**  We analyze the contributions of various model components to its performance. Figure 5 delineates the impact of components: NODE is a free-form second-order neural ODE, Adv corresponds to the advection ODE form, Att adds the attention in addition to convolutions, and ClimODE adds also the emission component. All components bring performance improvements, with the advection and emission model having the largest, and attention the least effect. More details are in Appendix E.

## 6  CONCLUSION AND FUTURE WORK

We present ClimODE, a novel climate and weather modeling approach implementing weather continuity. ClimODE precisely forecasts global and regional weather and also provides uncertainty quantification. While our methodology is grounded in scientific principles, it is essential to acknowledge its inherent limitations when applied to climate and weather predictions in the context of climate change. The historical record attests to the dynamic nature of Earth's climate, yet it remains uncertain whether ClimODE can reliably forecast weather patterns amidst the profound and unpredictable climate changes anticipated in the coming decades. Addressing this formidable challenge and also extending our method on newly curated global datasets (Rasp et al., 2023) represents a compelling avenue for future research.

## ACKNOWLEDGEMENTS

We thank the researchers at ECMWF for their open data sharing and maintenance of the ERA5 dataset, without which this work would not have been possible. We acknowledge CSC – IT Center for Science, Finland, for providing generous computational resources. This work has been supported by the Research Council of Finland under the *HEALED* project (grant 13342077).

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

## A  ETHICAL STATEMENT

Deep learning surrogate models have the potential to revolutionize weather and climate modeling by providing efficient alternatives to computationally intensive simulations. These advancements hold promise for applications such as nowcasting, extreme event predictions, and enhanced climate projections, offering potential benefits like reduced carbon emissions and improved disaster preparedness while deepening our understanding of our planet.

## B  DATA

We trained our model using the preprocessed version of ERA5 from WeatherBench (Rasp et al., 2020). It is a standard benchmark data and evaluation framework for comparing data-driven weather forecasting models. WeatherBench regridded the original ERA5 at 0.25° to three lower resolutions: 5.625°, 2.8125°, and 1.40625°. We utilize the 5.625° resolution dataset for our method and all other competing methods. See https://confluence.ecmwf.int/display/CKB/ERA5%3A+data+documentation for more details on the raw ERA5 data and Table 3 summarizes the variables used.

Table 3: ECMWF data variables in our dataset. *Static* variables are time-independent, *Single* represents surface-level variables, and *Atmospheric* represents time-varying atmospheric properties at chosen altitudes.

| Type | Variable name | Abbrev. | ECMWF ID | Levels |
|---|---|---|---|---|
| Static | Land-sea mask | lsm | 172 | |
| Static | Orography | | | |
| Single | 2 metre temperature | t2m | 167 | |
| Single | 10 metre U wind component | u10 | 165 | |
| Single | 10 metre V wind component | v10 | 166 | |
| Atmospheric | Geopotential | z | 129 | 500 |
| Atmospheric | Temperature | t | 130 | 850 |

### B.1  SPHERICAL GEOMETRY

We model the data in a 2D latitude-longitude grid $\Omega$, but take the earth geometry into account by considering circular convolutions at the horizontal borders (international date line), and reflective convolutions at the vertical boundaries (north and south poles). We limit the data to latitudes $\pm 88°$ to avoid the grid rows collapsing to the poles at $\pm 90°$.

## C  IMPLEMENTATION DETAILS

### C.1  MODEL-HYPERPARAMETERS

### C.2  ATTENTION CONVOLUTIONAL NETWORK

We include an attention convolutional network $f_{\text{att}}$ which captures *global* information by considering states across the entire Earth, enabling the modeling of long-distance connections. This attention network is structured around Key-Query-Value dot product attention, with Key, Query, and Value maps parameterized as convolutional neural networks as,

Table 4: Default hyperparameters for the emission model $g$

| Hyperparameter | Meaning | Value |
|---|---|---|
| Padding size | Padding size of each convolution layer | 1 |
| Padding type | Padding mode of each convolution layer | X: Circular, Y: Reflection |
| Kernel size | Kernel size of each convolution layer | 3 |
| Stride | Stride of each convolution layer | 1 |
| Residual blocks | Number of residual blocks | [3,2,2] |
| Hidden dimension | Number of output channels of each residual block | $[128, 64, \text{out channels}]$ |
| Dropout | Dropout rate | 0.1 |

Table 5: Default hyperparameters for the convolution network $f_{\text{conv}}$

| Hyperparameter | Meaning | Value |
|---|---|---|
| Padding size | Padding size of each convolution layer | 1 |
| Padding type | Padding mode of each convolution layer | X: Circular, Y: Reflection |
| Kernel size | Kernel size of each convolution layer | 3 |
| Stride | Stride of each convolution layer | 1 |
| Residual blocks | Number of residual blocks | [5,3,2] |
| Hidden dimension | Number of output channels of each residual block | $[128, 64, \text{out channels}]$ |
| Dropout | Dropout rate | 0.1 |

- **Key (K), Value (V)**: Key and Value maps are parameterized as 2-layer convolutional neural networks with stride=2 and $C_{K,V}$ as the latent embedding size. Based on the stride, this embeds every 4th pixel into a key, value latent vector of size $C_{K,V}$. We collect all embeddings into one tensor.
- **Query (Q)**: Query map is parametrized as 2-layer convolutional neural networks with stride=1 and $C_Q$ as the latent embedding size. This incorporates somewhat local information and embeds into $C_Q$ latent vector. We collect all embeddings into one tensor.

We compute the attention maps via dot-product maps as,

$$\beta = \texttt{softmax}(QK^\top)V \tag{14}$$

We consider a post-attention map for the attention coefficients as a 1-layer convolutional network with $1 \times 1$ filter size to map the latent vectors into output channels.

### C.3 METRICS

We assess benchmarks using latitude-weighted RMSE and Anomaly Correlation Coefficient (ACC) following the de-normalization of predictions.

$$\text{RMSE} = \frac{1}{N}\sum_t^N \sqrt{\frac{1}{HW}\sum_h^H\sum_w^W \alpha(h)(y_{thw} - u_{thw})^2}, \ \text{ACC} = \frac{\sum_{t,h,w}\alpha(h)\tilde{y}_{thw}\tilde{u}_{thw}}{\sqrt{\sum_{t,h,w}\alpha(h)\tilde{y}_{thw}^2 \sum_{t,h,w}\alpha(h)\tilde{u}_{thw}^2}} \tag{15}$$

where $\alpha(h) = \cos(h)/\frac{1}{H}\sum_{h'}^H \cos(h')$ is the latitude weight and $\tilde{y} = y - C$ and $\tilde{u} = u - C$ are averaged against empirical mean $C = \frac{1}{N}\sum_t y_{thw}$. The anomaly correlation coefficient (ACC) gauges a model's

ability to predict deviations from normal conditions. Higher ACC values signify better prediction accuracy, while lower values indicate poorer performance. It's a vital tool in meteorology and climate science for evaluating a model's skill in capturing unusual weather or climate events, aiding in forecasting system assessments. Latitude-weighted RMSE measures the accuracy of a model's predictions while considering the Earth's curvature. The weightage by latitude accounts for the changing area represented by grid cells at different latitudes, ensuring that errors in climate or spatial data are appropriately assessed. Lower latitude-weighted RMSE values indicate better model performance in capturing spatial or climate patterns.

## D TRAINING DETAILS

### D.1 DATA NORMALIZATION

We utilize 6-hourly forecasting data points from the ERA5 dataset and considered $K = 5$ quantities from the ERA5 dataset: ground temperature (t2m), atmospheric temperature (t), geopotential (z), and ground wind vector (u10, v10) and normalize the variables to $[0, 1]$ via min-max scaling. We use ten years of training data (2006–15), 2016 as validation data, and 2017–18 as testing data. There are 1460 data points per year and 2048 spatial points.

### D.2 DATA BATCHING

In our experiments, we utilize $K = 5$ quantities (See Appendix B) and spatial discretization of the earth to resolution $(H, W) = (32, 64)$ resulting in a total of $3KWH = 30720$ scalar ODEs. This can seem daunting, but they all *share the same differential function $f_\theta$*, that is, the time evolution at Tokyo and New York follows the same rules. The system can then be batched into a single image $\text{stack}[\mathbf{u}(t); \mathbf{v}(t); \psi]$ of size $(3K + C, H, W)$, which is input to $f_\theta(\cdot) : \mathbb{R}^{3K+C \times H \times W} \to \mathbb{R}^{3K \times H \times W}$ and can be solved in one forward pass.

$$\begin{bmatrix} \mathbf{u} \\ \mathbf{v} \end{bmatrix}(t) \in \mathbb{R}^{(3K+C) \times H \times W} \quad \begin{bmatrix} \dot{\mathbf{u}} \\ \mathbf{v} \end{bmatrix}(t) = \begin{bmatrix} \text{advection} \\ f_\theta \end{bmatrix} \in \mathbb{R}^{3K \times H \times W} \tag{16}$$

We batch the data points wrt to years, giving the batch of shape $(N \times B \times (3K + C) \times H \times W)$, where $B$ is the batch size and $N$ here denotes the number of years. We used batch-size $B = 8$ to train our model.

### D.3 OPTIMIZATION

We used Cosine-Annealing-LR[2] scheduler for the learning rate and also for the variance weight $\lambda_\sigma$ for L2 norm shown in Fig. 7 in the loss in Eq. 12. We trained our model for 300 epochs, and the scheduler variation is shown below.

### D.4 SOFTWARE AND HARDWARE

The model is implemented in PyTorch (Paszke et al., 2019) utilizing torchdiffeq (Chen et al., 2018) to manage our data and model training. We use euler as our ODE-solver that solves the dynamical system forward with a time resolution of 1 hour. The whole model training and inference is conducted on a single 32GB NVIDIA V100 device.

---

[2]https://pytorch.org/docs/stable/generated/torch.optim.lr_scheduler.CosineAnnealingLR.html

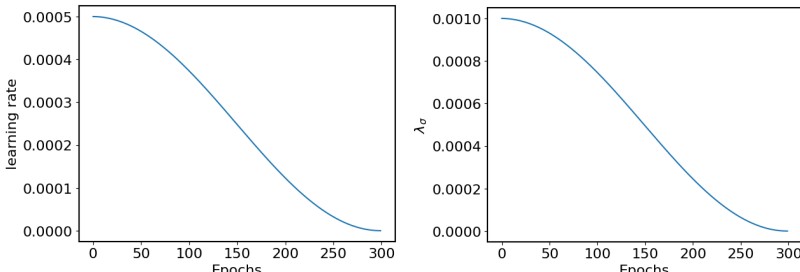

Figure 7: Learning rate and $\lambda_\sigma$ schedule wrt epochs.

### D.5 INITIAL VELOCITY INFERENCE

The neural transport model necessitates an initial velocity estimate, $\hat{\mathbf{v}}_k(\mathbf{x}, t_0)$, to initiate the ODE system (5). We estimate the missing velocity directly, $\mathbf{v}$, as a preprocessing step, for location $\mathbf{x}$, time $t$ and quantity $k$ to match the advection equation by penalized least-squares, where $\dot{u}$ is approximated by examining previous states $u(t < t_0)$ to obtain a numerical estimate of the change at $t_0$,

$$\hat{\mathbf{v}}_k(t) = \arg\min_{\mathbf{v}_k(t)} \left\{ \left|\left|\tilde{\dot{u}}_k(t) + \mathbf{v}_k(t) \cdot \tilde{\nabla}u_k(t) + u_k(t)\tilde{\nabla} \cdot \mathbf{v}_k(\mathbf{x}, t)\right|\right|_2^2 + \alpha \left|\left|\mathbf{v}_k(t)\right|\right|_{\mathbf{K}} \right\}, \quad (17)$$

where $\{\tilde{\cdot}, \tilde{\nabla}\}$ are numerical derivatives over time or space. We compute $\tilde{\dot{u}}_k(t)$ by utilizing torchcubicspline[3] to fit $\{\mathbf{u}_k(t-2), \mathbf{u}_k(t-1), \mathbf{u}_k(t)\}$ to get a smooth derivative approximation. The spatial gradients $\tilde{\nabla}$ are calculated using `torch.gradient` function of PyTorch. We additionally place a Gaussian zero-mean prior $\mathcal{N}(\text{vec } \mathbf{v}_k | \mathbf{0}, \mathbf{K})$ with a Gaussian RBF kernel $\mathbf{K}_{ij} = \text{rbf}(\mathbf{x}_i, \mathbf{x}_j)$ that results in spatially smooth initial velocities with smoothing coefficient $\alpha$. The distance for the $\text{rbf}(\mathbf{x}_i, \mathbf{x}_j)$ is computed as the euclidean norm between $\mathbf{x}_i$ and $\mathbf{x}_j$. This is optimized separately for each location $\mathbf{x}$ of the initial time $t_0$. We use Adam optimizer with a learning rate of 2 for 200 epochs. To get a balance between smoothing, local and global pattern we set the smoothing coefficient $\alpha = 10^{-7}$.

### E  ABLATION STUDY COMPONENTS

We conducted an extensive analysis to evaluate the individual contributions of each model component to its overall performance, as illustrated in Fig. 5. We delineate the impact of different components as,

- **NODE**: A basic second-order neural differential equation as, here $f_{\text{conv}}$ is parametrized by ResNet with the same set of parameters shown in Table 5,

$$\dot{u}_k(\mathbf{x}, t) = \mathbf{v}_k(\mathbf{x}, t) \quad (18)$$

$$\dot{\mathbf{v}}_k(\mathbf{x}, t) = f_{\text{conv}}\Big(\mathbf{u}(t), \nabla\mathbf{u}(t), \mathbf{v}(t), \psi\Big) \quad (19)$$

- **NODE+Adv**: This combines the second-order neural differential equation with the advection component, where $f_{\text{conv}}$ is parametrized by ResNet with the same set of parameters shown in Table 5,

$$\dot{u}_k(\mathbf{x}, t) = -\mathbf{v}_k(\mathbf{x}, t) \cdot \nabla u_k(\mathbf{x}, t) - u_k(\mathbf{x}, t)\nabla \cdot \mathbf{v}_k(\mathbf{x}, t) \quad (20)$$

$$\dot{\mathbf{v}}_k(\mathbf{x}, t) = f_{\text{conv}}\Big(\mathbf{u}(t), \nabla\mathbf{u}(t), \mathbf{v}(t), \psi\Big) \quad (21)$$

---

[3] https://github.com/patrick-kidger/torchcubicspline

- **NODE+Adv+Att**: This the `NODE+Adv` with the attention convolutional network to model both local and global effects, where $f_{\text{conv,att}}$ is parametrized by ResNet with the same set of parameters shown in Table 5 and Section C.2,

$$\dot{u}_k(\mathbf{x}, t) = -\mathbf{v}_k(\mathbf{x}, t) \cdot \nabla u_k(\mathbf{x}, t) - u_k(\mathbf{x}, t) \nabla \cdot \mathbf{v}_k(\mathbf{x}, t) \tag{22}$$

$$\dot{\mathbf{v}}_k(\mathbf{x}, t) = f_{\text{conv}}\Big(\mathbf{u}(t), \nabla \mathbf{u}(t), \mathbf{v}(t), \psi\Big) + \gamma f_{\text{att}}\Big(\mathbf{u}(t), \nabla \mathbf{u}(t), \mathbf{v}(t), \psi\Big) \tag{23}$$

ClimODE encompasses all the previous components with the emission model component, including the bias and variance components. The `NODE, NODE+Adv, NODE+Adv+Att` is trained by minimizing the MSE between predicted and truth observation as they output a point prediction and do not estimate uncertainty in the prediction. We employ unweighted RMSE as our evaluation metric to compare these methods. Our findings reveal a discernible hierarchy of performance improvement by incorporating each component, underscoring the vital role played by each facet in enhancing the model's downstream performance.

The Fig. 8 showcases the effect of emission model in modeling diurnal cycle and also predicting uncetainty.

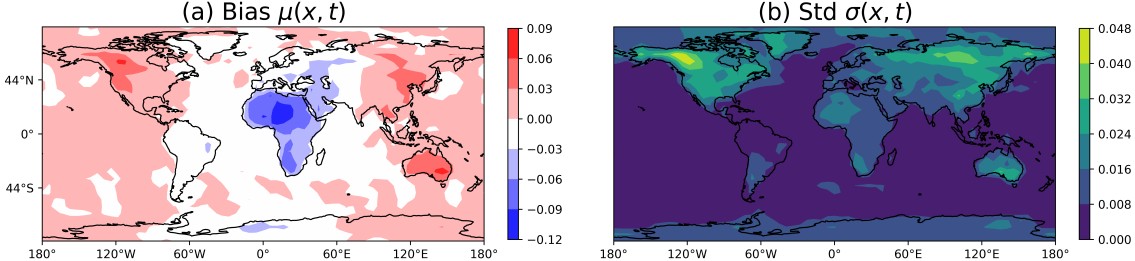

Figure 8: **Effect of emission model**: Global bias and standard deviation maps at 12:00 AM UTC. The bias explains day-night cycle **(a)**, while uncertainty is highest on land, and in north **(b)**.

## F    RESULTS SUMMARY

## G    LONGER HORIZON PREDICTIONS

Table 7 showcases the comparison of our method with ClimaX for for 72 hours (3 days) and 144 hours (6 days) lead time on latitude weighted RMSE and ACC metrics. We observe that the temperature and potential (t,t2m,z) are relatively stable over longer forecasts, while the wind direction (u10,v10) becomes unreliable over a long time, which is an expected result. ClimaX is also remarkably stable over long predictions but has lower performance. We see that our method achieve better performance as compared to ClimaX for longer horizon predictions.

## H    VALIDITY OF MASS CONSERVATION

To empirically study this, we analyzed how our current model retains the mass-conservation assumption and computed the integrals $I_{k,t} = \int u_k(\mathbf{x}, t) d\mathbf{x}$ over time and quantities. We discovered that the value is constant over time up to $10^{-12}$.

Table 6: Latitude weighted RMSE(↓) and ACC(↑) comparison with baselines on global forecasting on ERA5 dataset.

| Variable | Lead-Time (hours) | RMSE(↓) | | | | | ACC(↑) | | | | |
|---|---|---|---|---|---|---|---|---|---|---|---|
| | | NODE | ClimaX | FCN | IFS | ClimODE | NODE | ClimaX | FCN | IFS | ClimODE |
| z | 6 | 300.64 | 247.5 | 149.4 | 26.9 | 102.9 ±9.3 | 0.96 | 0.97 | 0.99 | 1.00 | 0.99 |
| | 12 | 460.23 | 265.3 | 217.8 | (N/A) | 134.8 ± 12.3 | 0.88 | 0.96 | 0.99 | (N/A) | 0.99 |
| | 18 | 627.65 | 319.8 | 275.0 | (N/A) | 162.7 ± 14.4 | 0.79 | 0.95 | 0.99 | (N/A) | 0.98 |
| | 24 | 877.82 | 364.9 | 333.0 | 51.0 | 193.4 ± 16.3 | 0.70 | 0.93 | 0.99 | 1.00 | 0.98 |
| | 36 | 1028.20 | 455.0 | 449.0 | (N/A) | 259.6 ± 22.3 | 0.55 | 0.89 | 0.99 | (N/A) | 0.96 |
| t | 6 | 1.82 | 1.64 | 1.18 | 0.69 | 1.16 ± 0.06 | 0.94 | 0.94 | 0.99 | 0.99 | 0.97 |
| | 12 | 2.32 | 1.77 | 1.47 | (N/A) | 1.32 ± 0.13 | 0.85 | 0.93 | 0.99 | (N/A) | 0.96 |
| | 18 | 2.93 | 1.93 | 1.65 | (N/A) | 1.47 ± 0.16 | 0.77 | 0.92 | 0.99 | (N/A) | 0.96 |
| | 24 | 3.35 | 2.17 | 1.83 | 0.87 | 1.55 ± 0.18 | 0.72 | 0.90 | 0.99 | 0.99 | 0.95 |
| | 36 | 4.13 | 2.49 | 2.21 | (N/A) | 1.75 ± 0.26 | 0.58 | 0.86 | 0.99 | (N/A) | 0.94 |
| t2m | 6 | 2.72 | 2.02 | 1.28 | 0.97 | 1.21 ± 0.09 | 0.82 | 0.92 | 0.99 | 0.99 | 0.97 |
| | 12 | 3.16 | 2.26 | 1.48 | (N/A) | 1.45 ± 0.10 | 0.68 | 0.90 | 0.99 | (N/A) | 0.96 |
| | 18 | 3.45 | 2.45 | 1.61 | (N/A) | 1.43 ± 0.09 | 0.69 | 0.88 | 0.99 | (N/A) | 0.96 |
| | 24 | 3.86 | 2.37 | 1.68 | 1.02 | 1.40 ± 0.09 | 0.79 | 0.89 | 0.99 | 0.99 | 0.96 |
| | 36 | 4.17 | 2.87 | 1.90 | (N/A) | 1.70 ± 0.15 | 0.49 | 0.83 | 0.99 | (N/A) | 0.94 |
| u10 | 6 | 2.3 | 1.58 | 1.47 | 0.80 | 1.41 ± 0.07 | 0.85 | 0.92 | 0.95 | 0.98 | 0.91 |
| | 12 | 3.13 | 1.96 | 1.89 | (N/A) | 1.81 ± 0.09 | 0.70 | 0.88 | 0.93 | (N/A) | 0.89 |
| | 18 | 3.41 | 2.24 | 2.05 | (N/A) | 1.97 ± 0.11 | 0.58 | 0.84 | 0.91 | (N/A) | 0.88 |
| | 24 | 4.1 | 2.49 | 2.33 | 1.11 | 2.01 ± 0.10 | 0.50 | 0.80 | 0.89 | 0.97 | 0.87 |
| | 36 | 4.68 | 2.98 | 2.87 | (N/A) | 2.25 ± 0.18 | 0.35 | 0.69 | 0.85 | (N/A) | 0.83 |
| v10 | 6 | 2.58 | 1.60 | 1.54 | 0.94 | 1.53 ± 0.08 | 0.81 | 0.92 | 0.94 | 0.98 | 0.92 |
| | 12 | 3.19 | 1.97 | 1.81 | (N/A) | 1.81 ± 0.12 | 0.61 | 0.88 | 0.91 | (N/A) | 0.89 |
| | 18 | 3.58 | 2.26 | 2.11 | (N/A) | 1.96 ± 0.16 | 0.46 | 0.83 | 0.86 | (N/A) | 0.88 |
| | 24 | 4.07 | 2.48 | 2.39 | 1.33 | 2.04 ± 0.10 | 0.35 | 0.80 | 0.83 | 0.97 | 0.86 |
| | 36 | 4.52 | 2.98 | 2.95 | (N/A) | 2.29 ± 0.24 | 0.29 | 0.69 | 0.75 | (N/A) | 0.83 |

Table 7: **Longer lead time predictions**: Latitude weighted RMSE(↓) and ACC(↑) for longer lead times in global forecasting using the ERA5 dataset, in comparison with ClimaX.

| Variable | Lead-Time (hours) | RMSE(↓) | | ACC(↑) | |
|---|---|---|---|---|---|
| | | ClimaX | ClimODE | ClimaX | ClimODE |
| z | 72 | 687.0 | 478.7 ±48.3 | 0.73 | 0.88 ±0.04 |
| | 144 | 801.9 | 783.6 ±37.3 | 0.58 | 0.61 ±0.13 |
| t | 72 | 3.17 | 2.58 ±0.16 | 0.76 | 0.85 ±0.06 |
| | 144 | 3.97 | 3.62 ±0.21 | 0.69 | 0.77 ±0.16 |
| t2m | 72 | 2.87 | 2.75 ±0.49 | 0.83 | 0.85 ±0.14 |
| | 144 | 3.38 | 3.30 ±0.23 | 0.83 | 0.79 ±0.25 |
| u10 | 72 | 3.70 | 3.19±0.18 | 0.45 | 0.66 ±0.04 |
| | 144 | 4.24 | 4.02 ±0.12 | 0.30 | 0.35 ±0.08 |
| v10 | 72 | 3.80 | 3.30 ±0.22 | 0.39 | 0.63 ±0.05 |
| | 144 | 4.42 | 4.24±0.10 | 0.25 | 0.32 ±0.11 |

# I    CRPS (CONTINUOUS RANKED PROBABILITY SCORE) AND CLIMATE FORECASTING

We further assessed our model using CRPS (Continuous Ranked Probability Score), as depicted in Figure 10. This analysis highlights our model's proficiency in capturing the underlying dynamics, evident in its accurate prediction of both mean and variance.

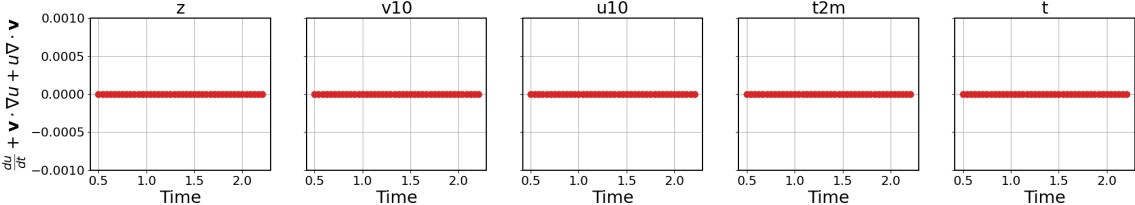

Figure 9: Validity of the mass conservation assumption of the ODE.

To showcase the effectiveness of our model in climate forecasting, we predicted average values over a one-month duration for key meteorological variables sourced from the ERA5 dataset: ground temperature (t2m), atmospheric temperature (t), geopotential (z), and ground wind vector (u10, v10). Employing identical data-preprocessing steps, normalization, and model hyperparameters as detailed in previous experiments, Figure 10 illustrates the performance of ClimODE compared to FourCastNet in climate forecasting. Particularly noteworthy is our method's superior performance over FourCastNet at longer lead times, underscoring the multi-faceted efficacy of our approach.

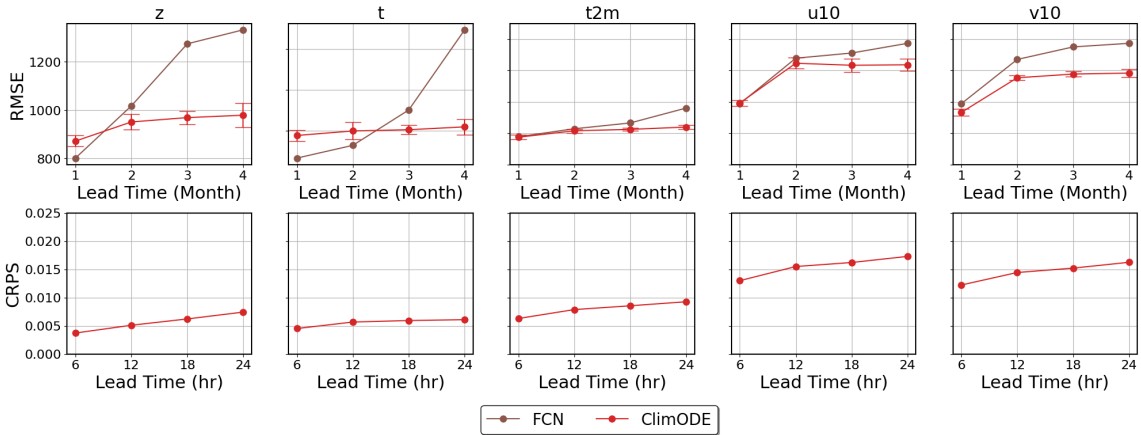

Figure 10: **CRPS and Monthly Forecasting**: RMSE($\downarrow$) comparison with FourCastNet (FCN) for monthly forecasting and CRPS scores for ClimODE.

## J    CORRELATION PLOTS

To demonstrate the emerging couplings of quantities (ie. wind, temperature, pressure potential), we below plot the emission model $\mathbf{u}^{\text{pred}}(\mathbf{x}, t) \in \mathbb{R}^5$ pairwise densities averaged over space $\mathbf{x}$ and time $t$. These effectively capture the correlations between quantities in the simulated weather states. These show that temperatures (t,t2m) and potential (z) are highly correlated and bimodal; the horizontal and vertical wind direction are independent (u10,v10); and there is little dependency between the two groups. These plots indicate that the emission model is highly aligned with data and does not indicate any immediate biases or skews. These results are averaged over space and time, and spatially local variations are still possible. The mean $\mu$ plots show that means match data well. The standard deviation $\sigma$ plots show some bimodality of predictions with either no or moderate uncertainty.

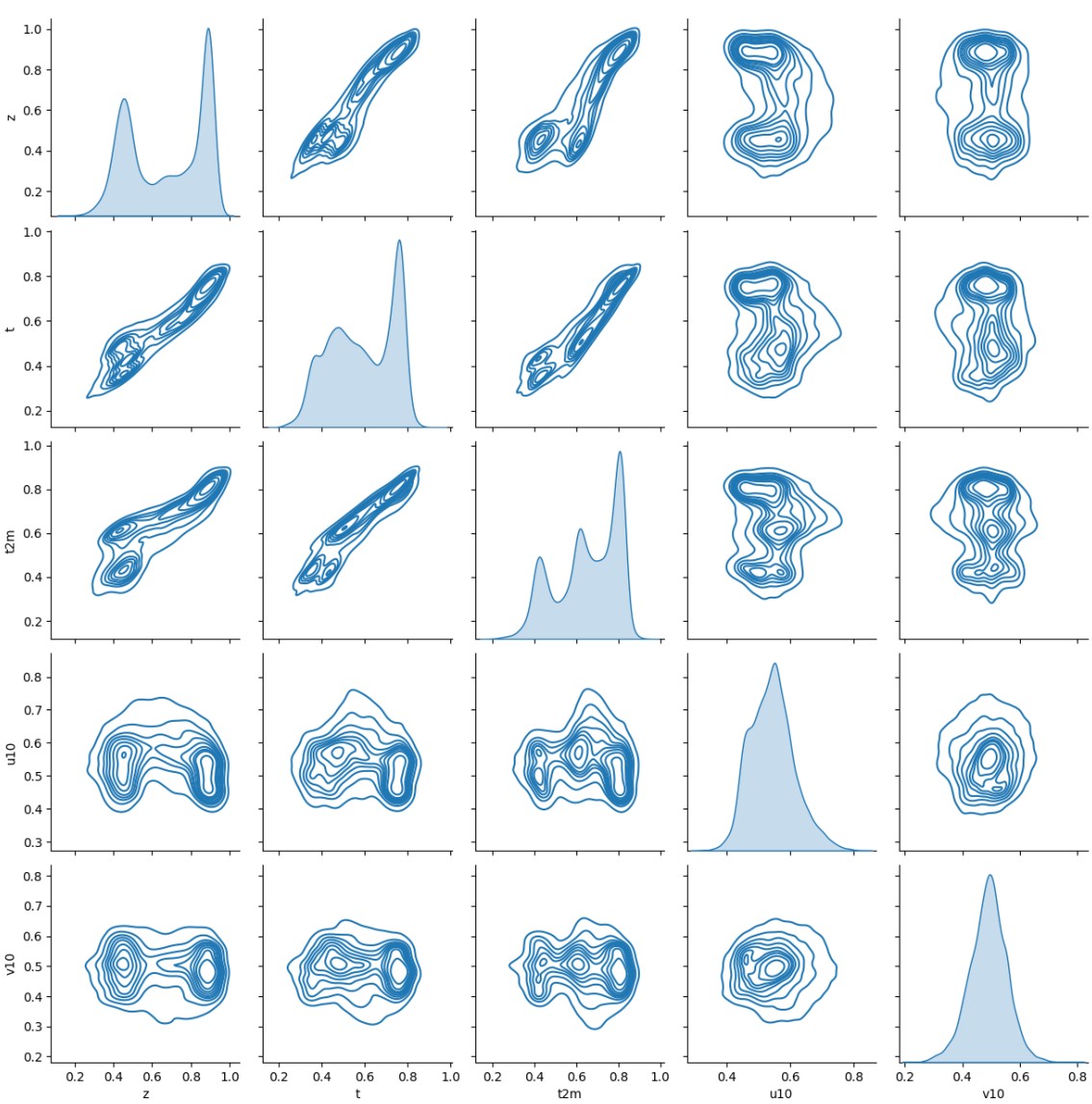

Figure 11: Pairwise correlation among the predicted variables by the model.

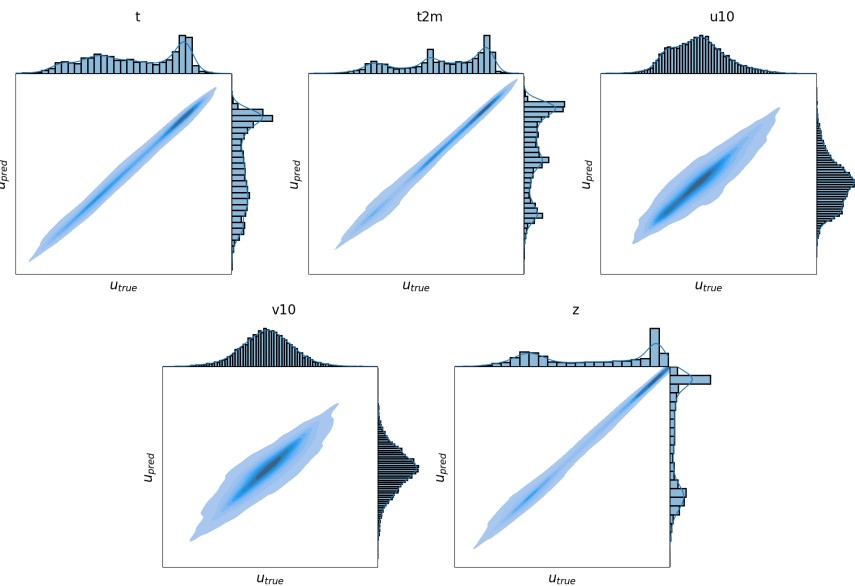

Figure 12: Correlation between $u_{pred}$ and $u_{true}$ for different observables, showing the efficacy of our model to predict the observables accurately.

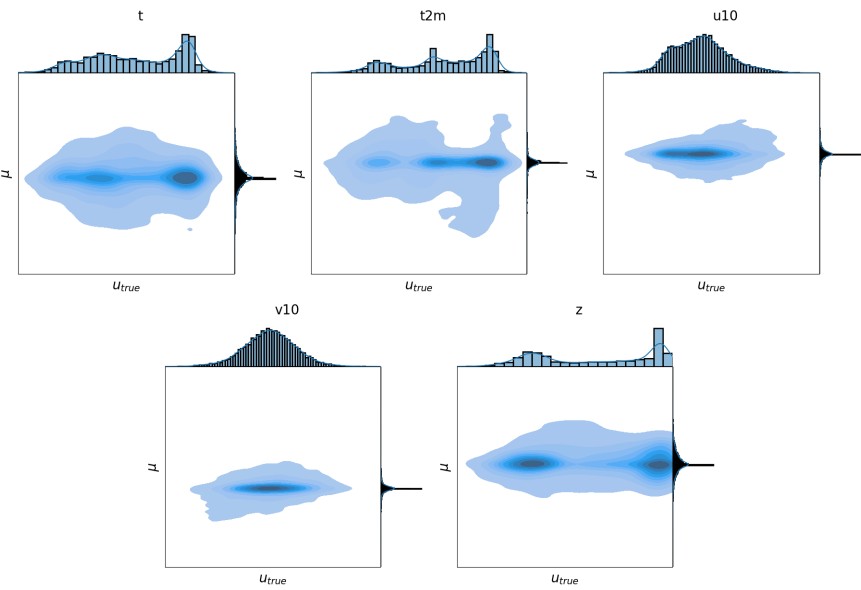

Figure 13: Correlation between $\mu$ and $u_{true}$ for different observables.

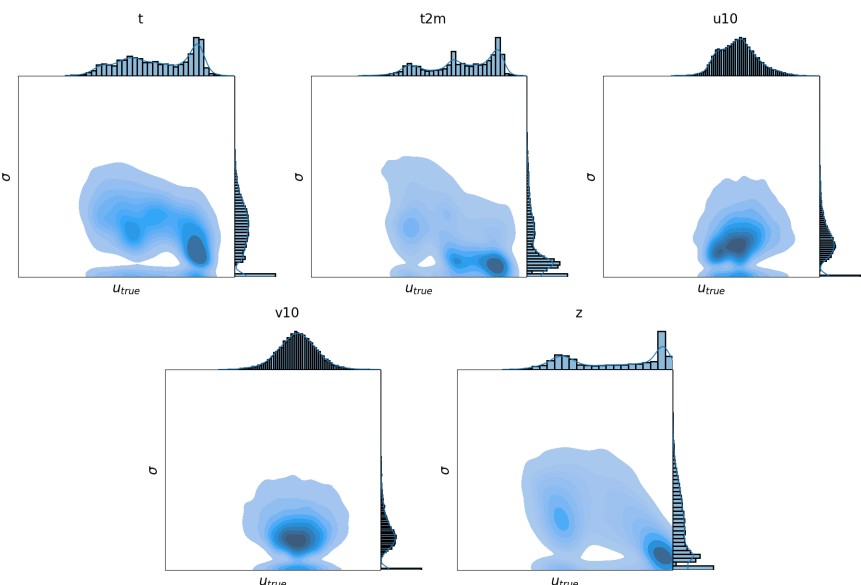

Figure 14: Correlation between $\sigma$ and $u_{true}$ for different observables.

