# OpenReview forum: "ClimODE: Climate and Weather Forecasting with Physics-informed Neural ODEs"
_ICLR.cc/2024/Conference — ICLR 2024 oral_

### Official Review · Reviewer_MKLC · 2023-10-17

**Soundness:** 4 excellent
**Presentation:** 3 good
**Contribution:** 3 good
**Rating:** 8
**Confidence:** 4

**Summary:**

ClimODE aims to improve climate prediction by integrating physical principles into a neural ordinary differential equation (ODE) framework. Unlike traditional black-box models, ClimODE incorporates the concept of advection from statistical mechanics, ensuring that the model considers the spatial movement of weather quantities over time. The model employs neural ODEs to model the weather evolution with value-conserving dynamics. The architecture includes components for local convolutions and global attention, allowing it to capture both local and global weather influences. ClimODE also addresses uncertainty in predictions and source variations through a probabilistic emission model. This feature allows the model to quantify prediction uncertainties and adapt to various source variations. ClimODE is reported to outperform existing data-driven methods in global and regional forecasting.

**Strengths:**

1. Physics-Informed Modeling: ClimODE incorporates physical principles, ensuring that the model aligns with our first-principle understanding of the meteorological dynamics. ClimODE follows the partial differential continuity equation and solves a latent equation in such a form, and this makes both the physical insights and model training easier to understand and follow. Because of the use of Neural ODE, the prediction would be continuous in spacetime. I like the cute and nice idea.

2. Model Design: The use of both local convolutions and global attention allows the model to capture a wide range of influences on weather patterns, enhancing its predictive capabilities with both locality inductive bias and long-term interaction capability. The model’s ability to quantify uncertainty and/or model source variation is an additional advantage, making its forecasts more reliable and comprehensive (like a latent force). The spatio-temporal embedding also gives the model extra power to reflect the geographical differences with spatial-temporal variation, which makes a lot of sense.

3. Performance and Efficiency: ClimODE achieves superior performance over some existing deep learning weather forecasting models, with an order of magnitude smaller parameterization, making it computationally efficient.

4. Presentation: Very clear description of initial condition modeling, PDE to ODE modeling, Advection and Flow Velocity modeling, etc.

**Weaknesses:**

1. Limited Performance and Comparison: The proposed model, although surpasses vanilla Neural ODE and ClimaX, still cannot compete with IFS (ECMWF NWP), let alone bigger models such as FourCastNet and Pangu-Weather that have already reported better results than NWP. There lacks enough comparison to other models in general, such as Adaptive Fourier Neural Operator (practiced by FourcastNet), previous weather forecasting methods like [1, 2], even models for similar tasks such as spatio-temporal traffic forecasting or video forecasting, etc. Comparison to Neural ODE is more of an ablation study and ClimaX is a weak baseline marginally better than ResNet. Comparing only with ClimaX is definitely not convincing enough.

2. Limited Physical Complexity: The partial differential continuity equation is indeed a fundamental concept in fluid dynamics as it describes how quantities such as mass, moisture, or energy are transported and conserved in the atmosphere over time and space. What makes it less predictive in the real world is that this equation comes with assumptions such as the homogeneity and isotropy of the fluid. In cases where these assumptions are violated, the equation might not hold perfectly. I am not certain whether relying too much on this physical equation would be optimal for modeling the complex dynamics of weather, especially in finer-granular resolution with more weather factors and potentially larger noises.

3. Limited Physical Understanding: In all, the partial differential continuity equation is a general-form conservation equation rather than any specific equations explaining each weather factor. Other than that, because neural ODE is essentially still a black-box model due to using neural network, it remains hard to know, for example, the relationships and the interactions between different weather variables.

[1] Yan Han, Lihua Mi, Lian Shen, CS Cai, Yuchen Liu, Kai Li, and Guoji Xu. A short-term wind speed prediction method utilizing novel hybrid deep learning algorithms to correct numerical weather forecasting. Applied Energy, 312:118777, 2022.
[2] Xiaoying Yang, Shuai Yang, Mou Leong Tan, Hengyang Pan, Hongliang Zhang, Guoqing Wang, Ruimin He, and Zimeng Wang. Correcting the bias of daily satellite precipitation estimates in tropical regions using deep neural network. Journal of Hydrology, 608:127656, 2022.

**Questions:**

If the several ideas behind ClimODE is effective as suggested by the ablation study, I would really be interested to see its performance in a large model compared to Pangu or FourcastNet, for example. Those benchmarks are exceptional since they excel NWP results in many cases and have been tested/used even for recent disaster predictions. It is just like the LLM that, if it becomes powerful enough, everyone will be amazed and will use it. If the ultimate goal of this paper is to propose a strong SOTA model against existing models, it is strongly recommended to compare with those large models and consider adapting to this line in the future research.

Since the paper mentions efficiency, I wonder how does the model perform with more parameters. Would the performance increase? Or, it is more like a nice small model to save us from excessive computational burden?

Are there any reasons why ClimODE performs worse than ClimaX in t2m?

How does the model perform with long-term prediction, say, for 7 days or 14 days? The authors mention the emission source model as a strong inductive bias that prevents long-horizon forecast collapses and it seems to work based on the ablation study, so I wonder that.

---

> ### Author Response · Authors · 2023-11-16
> **Response**
>
> Thank you very much for many excellent suggestions! We have acted on all of these, and additionally, address all your questions and comments below.
>
> > Limited Performance and Comparison: The proposed model, although surpasses vanilla Neural ODE and ClimaX, still cannot compete with IFS (ECMWF NWP), let alone bigger models such as FourCastNet and Pangu-Weather that have already reported better results than NWP. There lacks enough comparison to other models in general, such as Adaptive Fourier Neural Operator (practiced by FourcastNet), previous weather forecasting methods like [1, 2], even models for similar tasks such as spatio-temporal traffic forecasting or video forecasting, etc. Comparison to Neural ODE is more of an ablation study and ClimaX is a weak baseline marginally better than ResNet. Comparing only with ClimaX is definitely not convincing enough.
>
> **Comparison with FourCastNet.** Thank you for the suggestion. We could not compare to PanguWeather (no public code, model files do not support our data resolution) or GraphCast (public repository seemed incomplete). However, we include an additional experiment compared to FourCastNet. To have comparable results, we re-trained the FourCastNet on our dataset and used their recommended training regime and hyperparameters.
>
> Our results reinforce the efficacy of ClimODE. We note that our method is more accurate in terms of RMSE (in fact, quite significantly especially for the geopolitical variable z) and largely competitive in terms of ACC. We will include these results in the paper.
>
> |          |                   | RMSE        |                        | ACC         |          |
> |----------|-------------------|-------------|------------------------|-------------|----------|
> | Variable | Lead Time (hours) | FourCastNet | ClimODE                | FourCastNet | ClimODE  |
> | z        | 6                 | 149.4       | **102.9**   $\pm$ 9.3 | 0.73        | 0.99     |
> |          | 12                | 217.8       | **134.8**   $\pm$ 12.3     | 0.58        | 0.99     |
> |          | 18                | 275.0       | **162.7**   $\pm$ 14.4     | 0.58        | 0.98     |
> |          | 24                | 333.0       | **193.4**   $\pm$ 16.3     | 0.58        | 0.98     |
> | t        | 6                | 1.18        | **1.16**   $\pm$ 0.06      | 0.99        | 0.97     |
> |          | 12               | 1.47        | **1.32**   $\pm$ 0.13      | 0.99        | 0.96     |
> |          | 18               | 1.65        | **1.47**   $\pm$ 0.16      | 0.99        | 0.96     |
> |          | 24               | 1.83        | **1.55**   $\pm$ 0.18      | 0.99        | 0.95     |
> | t2m        | 6                | 1.28        | **1.21**   $\pm$ 0.09      | 0.99        | 0.97     |
> |          | 12               | 1.48        | **1.45**   $\pm$ 0.10      | 0.99        | 0.96     |
> |          | 18               | 1.61        | **1.43**   $\pm$ 0.09      | 0.99        | 0.96     |
> |          | 24               | 1.68        | **1.40**   $\pm$ 0.09      | 0.99        | 0.96     |
> | u10        | 6                | 1.47        | **1.41**   $\pm$ 0.07      | 0.93        | 0.91     |
> |          | 12               | 1.89        | **1.81**   $\pm$ 0.09      | 0.91        | 0.89     |
> |          | 18               | 2.05        | **1.97**   $\pm$ 0.11      | 0.89        | 0.88     |
> |          | 24               | 2.33        | **2.01**   $\pm$ 0.10      | 0.89        | 0.87     |
> | v10        | 6                | 1.54        | **1.53**   $\pm$ 0.08      | 0.95        | 0.92     |
> |          | 12               | 1.81        | 1.81   $\pm$ 0.12      | 0.91        | 0.89     |
> |          | 18               | 2.11        | **1.96**   $\pm$ 0.16      | 0.88        | 0.88     |
> |          | 24               | 2.39        | **2.04**   $\pm$ 0.10      | 0.85        | 0.86     |

---

> > ### Author Response · Authors · 2023-11-16
> > **Response Continued**
> >
> > > Limited Physical Complexity: The partial differential continuity equation is indeed a fundamental concept in fluid dynamics as it describes how quantities such as mass, moisture, or energy are transported and conserved in the atmosphere over time and space. What makes it less predictive in the real world is that this equation comes with assumptions such as the homogeneity and isotropy of the fluid. In cases where these assumptions are violated, the equation might not hold perfectly. I am not certain whether relying too much on this physical equation would be optimal for modeling the complex dynamics of weather, especially in finer-granular resolution with more weather factors and potentially larger noises.
> >
> > These are all great points, and we agree. We believe that a performant climate model likely needs to strike a balance between data-driven deep learning (often black-box) practises, and theoretical modeling and understanding of the underlying physics. Earlier methods, such as PanguWeather or GraphCast, lean quite heavily on the deep learning side. In contrast, our contribution is to combine neural networks with advection equations, which makes our modeling significantly more principled over relying solely on deep learning. \\
> >
> > To our knowledge, our study is the first method that introduces principled continuous-time dynamics to the problem, and further research is needed to identify the extent of physics to be injected in the models. In our model the advection primarily contributes mass conservation, which we show empirically to be a very useful inductive bias.
> >
> > > Limited Physical Understanding: In all, the partial differential continuity equation is a general-form conservation equation rather than any specific equations explaining each weather factor. Other than that, because neural ODE is essentially still a black-box model due to using neural network, it remains hard to know, for example, the relationships and the interactions between different weather variables.
> >
> > Great remarks, and we again agree. The advection conservation equations have two roles in our study. On one hand, their foundation in physics allows us to interpret and analyse the proposed model from new perspectives (to the deep learning community), as well as  understand the model predictions in terms of divergence, transport and gradient fields (as shown in Figure 1 and 2). On the other hand, the induced `mass' conservation ensures that predictions do not lose or gain value, stabilising them and providing us with improved empirical accuracy. We are excited to study the physical modeling choices further in future, and experiment with diffusion, source and sink fields as well.
> >
> > The interdependency of climate variables is another incisive question. To demonstrate the couplings of quantities (i.e., wind, temperature, potential), we plot below the emission model  $\mathbf{u}^{\mathrm{pred}}(\mathbf{x},t) \in \mathbb{R}^5$ pairwise densities averaged over space $\mathbf{x}$ and time $t$. These effectively capture the correlations between quantities in the simulated weather states. These show that temperatures (t,t2m) and potential (z) are highly correlated and bimodal; the horizontal and vertical wind direction are independent (u10,v10); and there is little dependency between the two groups. We will include these plots in the appendix.
> >
> > Plot: https://postimg.cc/0McVK5r5
> >
> > > Since the paper mentions efficiency, I wonder how does the model perform with more parameters. Would the performance increase? Or, it is more like a nice small model to save us from excessive computational burden?
> >
> > This is a great suggestion, and an interesting question. Given the current dataset we suspect that more parameters likely would not result in significantly improved performance.
> >
> > That said, we are very interested in expanding the work to higher spatial resolutions, which surely would necessitate at least somewhat larger networks to be able to model the more fine-grained details. In principle, we expect that ClimODE could serve as a powerful and compelling foundation model for various climate prediction tasks by leveraging more data (e.g., historical data) and computation.

---

> > > ### Author Response · Authors · 2023-11-16
> > > **Response Continued**
> > >
> > > > If the several ideas behind ClimODE is effective as suggested by the ablation study, I would really be interested to see its performance in a large model compared to Pangu or FourcastNet, for example. Those benchmarks are exceptional since they excel NWP results in many cases and have been tested/used even for recent disaster predictions. It is just like the LLM that, if it becomes powerful enough, everyone will be amazed and will use it. If the ultimate goal of this paper is to propose a strong SOTA model against existing models, it is strongly recommended to compare with those large models and consider adapting to this line in the future research.
> > >
> > > We agree and are keen to compare these models as well. However, there are some practical hurdles: the large DL models have been trained with higher resolution data, and we cannot directly obtain comparable prediction trajectories due to different test data.
> > >
> > > These large weather models demonstrate significant scale, requiring substantial computational resources, extensive pre-training, and other prerequisites. These requirements impose practical limitations on the training process. We argue that there is also value in more compact models (in our case up to 100x less parameters) that allow rapid and more accessible development for the community.
> > >
> > > We would also like to emphasize that the primary goal of our work is to demonstrate the benefits of incorporating physical inductive biases in neural networks, and present a qualitatively improved model that has (i) solid foundations in physics, (ii) outputs principled uncertainty distributions, and (iii) is fundamentally continuous-time. We have also demonstrated good empirical performance. However, our primary goal is not to best other large climate foundations models in benchmarks, and we have chosen instead to focus on principled method developments and see perhaps in future these ideas be integrated into the large foundation models as well (indeed, recently NowCastNet incorporated autoregressive advection equations to its Transformer pipeline).
> > >
> > > > How does the model perform with long-term prediction, say, for 7 days or 14 days? The authors mention the emission source model as a strong inductive bias that prevents long-horizon forecast collapses and it seems to work based on the ablation study, so I wonder that.
> > >
> > > **Long-term prediction with ClimODE.** Great suggestion! We performed our main comparison with 6 days forecasts, and also include a 3 day reference values for comparison below. We see that our model accuracy drops relatively little over time. Also, ClimODE continues to perform better than ClimaX.
> > >
> > > |          |                   | RMSE        |                        | ACC         |          |
> > > |----------|-------------------|-------------|------------------------|-------------|----------|
> > > | Variable | Lead Time (hours)   | ClimaX | ClimODE | ClimaX | ClimODE |
> > > | z  | 72| 687.0  | 478.7   $\pm$ 48.3    | 0.73 | 0.88   |
> > > |    | 144| 801.9  | 783.6   $\pm$ 37.3   | 0.58  | 0.61   |
> > > | t  | 72| 3.17 | 2.58   $\pm$ 0.16    | 0.76 | 0.85   |
> > > |    | 144| 3.97  | 3.62   $\pm$ 0.21   | 0.69  | 0.77   |
> > > | t2m  | 72| 2.87 | 2.75   $\pm$ 0.49    | 0.83 | 0.85   |
> > > |    | 144| 3.38  | 3.30   $\pm$ 0.23   | 0.83  | 0.79   |
> > > | u10 | 72| 3.70 | 3.19   $\pm$ 0.18    | 0.45 | 0.66   |
> > > |    | 144| 4.24  | 4.02   $\pm$ 0.12   | 0.30  | 0.35   |
> > > | u10 | 72| 3.80 | 3.80   $\pm$ 0.22    | 0.39 | 0.63   |
> > > |    | 144| 4.42  | 4.42   $\pm$ 0.10   | 0.25  | 0.32   |
> > >
> > > We are grateful for your thoughtful review. We hope our response has addressed your questions and concerns, and will appreciate the same being reflected in your stronger support for this work.

---

> ### Comment · Reviewer_MKLC · 2023-11-16
> **Official Reply**
>
> Thanks for the update. Yes, I totally agree upon the motivation to incorporate physical mechanism. The comparison with FourcastNet seems very promising. Based on that, I would love to raise my score. It seems that when predicting steps increase, the performance drops to a level closer to ClimaX, so the long-term prediction could be less effective.

---

> > ### Author Response · Authors · 2023-11-23
> > **Response**
> >
> > Thank you so much for your response and support for this work. We are delighted that you share our enthusiasm for this work.
> > We are also glad to note that you find comparisons with FourcastNet very promising.
> >
> > Thank you for your observation about the effect of longer lead times on the performance gap. We would like to point out that ClimaX necessitates training for a designated lead time (according to the code and description provided in their repository) and operates non-iteratively, whereas we adopt an iterative approach that makes ClimODE significantly more flexible and amenable for practical scenarios.
> >
> > Once again, we gratefully acknowledge your help in reinforcing the merits of this work as well as your strong support for this paper.

---

### Official Review · Reviewer_THXC · 2023-10-24

**Soundness:** 3 good
**Presentation:** 4 excellent
**Contribution:** 4 excellent
**Rating:** 8
**Confidence:** 4

**Summary:**

The authors present ClimODE, a new method for weather forecasting that provides uncertainty esimates, benchmarked on ERA5 data from WeatherBench. This method is based on first principles from physics, namely the *continuity equation* which mathematically formulates that in a system, quantities are conserved (modulo any sources/sinks). Neural ODEs are used to model the continuity PDE and the flow velocity is parametrized using a neural net that has two components: A CNN that models local interactions and an attention mechanism that model long-range interactions. Furthermore, an emission model is added on top that serves to estimate first and second order moments of the true underlying solution (bias and uncertainty).

ClimODE greatly outperforms the baseline NODE which doesn't take into account the physical principles of advection nor does it correct its solution by the proposed emisison models, moreover, ClimODE also outperform ClimaX, one of the recent state-of-the-art weather forecasting models on the regridded 5.625 ° ERA5 data.

Finally, strong ablation study results are shown that indicate the important role of taking into account the underlying physical principles as well as the emission model.

**Strengths:**

- Proposed method that is based on first principles from physics and provides interpretability of the solution: i.e. explicit transport, compression and flow velocity terms.
- Provides efficient uncertainty estimation by learning the bias and the standard deviation, unlike previous methods that have to rely on ensembling which is computationally intensive.
- Simple model, but still manages to outperform ClimaX which would make this method not only a strong baseline for future work but also a method to quickly iterate on and improve.
- Strong ablation studies demonstrating the design choices made by ClimODE over NODE as well as the effect of the emission model.

**Weaknesses:**

- "Closed system assumption implies value-preserving manifold" is unclear and needs more justification. While the conclusion that: $$\mathbb{E}_x[u_k(\mathbf{x},t)]=\text{const.}$$

sounds intuitive, it's not justified rigourously. The expectation is taken with respect to which density? I would expect the quantity to be preserved would be $\int_{\Omega} u_k(\mathbf{x},t)dV$, since that's the total quantity over the whole "volume" you're considering which should be preserved (e.g. when $u_k=\rho$, the integral over the volume becomes the mass). In any case, since this is a Machine Learning submission, it would be better if this part is thoroughly justified.

- Benchmark only up to 36 hours while state-of-the-art methods reported results for up to 10 days ahead. The paper would be better if it reported results for at least five or seven days ahead.
- Authors mention that other deep learning methods lacked open-source code, while that's true for PanguWeather (who only provide a pseudo-code "implementation"), it's not for the others:
    - FourCastNet: https://github.com/NVlabs/FourCastNet
    - GraphCast: https://github.com/google-deepmind/graphcast

 Given that GraphCast seem to outperform ClimaX, it would have been good to compare against it as well and also FourCastNet since it's one of the first papers to perform weather forecasting on such a scale.
- Authors claim that the vaue-preserving manifold (that emanates from the closed system assumption), presents a strong inductive bias for long-term forecasts, yet, using Euler scheme to solve the ODE is known to be unstable and it'll especially not conserve the quantity we want preserved, so that inductive bias is no longer enforced. It would be better to include that limitation in the paper and acknowledge that while it is a strong inductive bias, it's hard to enforce in practice. This is further seen from Table 2 which shows that the error does increase dramatically with lead time and that suggests that ClimODE is better at inferring the true physics but not in mitigating the error propagation in long-term forecasts.
- While Figure 6 shows qualitatively the soundness of the predicted bias and variance, there's no quantitative approach that evaluates the quality of the bias and variance output by the model. A metric like CRPS (Continuous Ranked Probability Score) can showcase that.

**Questions:**

- Table 1 says that NowCastNet doesn't provide uncertainty estimation, but it's a generative model which can provide such estimates and in general approximate the true underlying distribution.
- In section 3.2, it's unclear why $\mathbf{\dot{v}}_k(\mathbf{x}, t) = \ddot{u}_k(x,t)$, especially when $\ddot{u}_k(x,t)$ is not a vector.
- In section 3.6, how is $\tilde{\dot{u}}(t_0)$ numerically apprixmated from past states?
- Why not use different time-resolutions for solving the ODE and assessing their effect? Same goes for the ODE-solver. Given that you state that Runge-Kutta can be used with a low computational cost, it would add more quality to the paper overall if you include it as well.
- How long does it take to train?
- Lacks training details for ClimaX as well as the training runtimes and number of GPUs used for ClimaX.

---

> ### Author Response · Authors · 2023-11-16
> **Response**
>
> Thanks so much for your thoughtful comments and excellent suggestions! We've acted on all of them, and also address all your concerns, as we describe below.
>
> > "Closed system assumption implies value-preserving manifold" is unclear and needs more justification. While the conclusion sounds intuitive, it's not justified rigorously. The expectation is taken with respect to which density? I would expect the quantity to be preserved would be $\int_{\Omega} u_{k}(x,t)dV$, since that's the total quantity over the whole "volume" you're considering which should be preserved (e.g. when the integral over the volume becomes the mass). In any case, since this is a Machine Learning submission, it would be better if this part is thoroughly justified.
>
> Thanks for pointing this out. Indeed, our expectation notation is imprecise, and we should have written a spatial integral $\int u_k(\mathbf{x},t) d\mathbf{x}$ as you suggest. We will fix this in the paper. We also analysed our model, and found out that the total spatial value is practically constant over time over the ODE forward solver (it shows a flat line over time). We will include this plot in the appendix.
>
> Plot: https://postimg.cc/w709xD9n
>
> > Benchmark only up to 36 hours while state-of-the-art methods reported results for up to 10 days ahead. The paper would be better if it reported results for at least five or seven days ahead.
>
> **Benchmarking with longer lead times.** Many thanks for the suggestion. Based on your feedback, we conducted additional experiments. We show below in the table the RMSE and ACC results for 6-day lead time, and also include 3-day prediction as a reference. We  also include ClimaX predictions for comparison. These results show that temperature and potential (t,t2m,z) are relatively stable over longer forecasts, while the wind direction (u10,v10) becomes unreliable over longer times. We note that while ClimaX is also remarkably stable over long predictions, it has lower performance compared to ClimODE. We will include this table in the appendix.
>
>
> |          |                   | RMSE        |                        | ACC         |          |
> |----------|-------------------|-------------|------------------------|-------------|----------|
> | Variable | Lead Time (hours)   | ClimaX | ClimODE | ClimaX | ClimODE |
> | z  | 72| 687.0  | 478.7   $\pm$ 48.3    | 0.73 | 0.88   |
> |    | 144| 801.9  | 783.6   $\pm$ 37.3   | 0.58  | 0.61   |
> | t  | 72| 3.17 | 2.58   $\pm$ 0.16    | 0.76 | 0.85   |
> |    | 144| 3.97  | 3.62   $\pm$ 0.21   | 0.69  | 0.77   |
> | t2m  | 72| 2.87 | 2.75   $\pm$ 0.49    | 0.83 | 0.85   |
> |    | 144| 3.38  | 3.30   $\pm$ 0.23   | 0.83  | 0.79   |
> | u10 | 72| 3.70 | 3.19   $\pm$ 0.18    | 0.45 | 0.66   |
> |    | 144| 4.24  | 4.02   $\pm$ 0.12   | 0.30  | 0.35   |
> | u10 | 72| 3.80 | 3.80   $\pm$ 0.22    | 0.39 | 0.63   |
> |    | 144| 4.42  | 4.42   $\pm$ 0.10   | 0.25  | 0.32   |
>
>
> > Authors claim that the value-preserving manifold (that emanates from the closed system assumption), presents a strong inductive bias for long-term forecasts, yet, using Euler scheme to solve the ODE is known to be unstable and it'll especially not conserve the quantity we want preserved, so that inductive bias is no longer enforced. It would be better to include that limitation in the paper and acknowledge that while it is a strong inductive bias, it's hard to enforce in practice. This is further seen from Table 2 which shows that the error does increase dramatically with lead time and that suggests that ClimODE is better at inferring the true physics but not in mitigating the error propagation in long-term forecasts.
>
> Thanks for pointing this out. To empirically study this, we analysed how our current model retains the mass-conservation assumption and computed the integrals $I_{k,t} = \int u_k(\mathbf{x},t)d\mathbf{x}$ over time and quantities. We found out that the value is constant over time up to order of $10^{-12}$. Plot link: https://postimg.cc/w709xD9n
>
> Thus it seems that our choice of second-order ODE with Euler with 1 hour increment is a good compromise in terms of accuracy and efficiency, but we agree that higher precision solvers should be preferred. We hypothesise that Euler is sufficient since our dynamics are relatively smooth (eg. Fig 6): the dynamics $\dot{u}_k$ largely models the spatiotemporal trends in climate, while we offload the day-night fluctuations to the instantaneous emission model.

---

> > ### Author Response · Authors · 2023-11-16
> > **Response Continued**
> >
> > > Authors mention that other deep learning methods lacked open-source code; while that's true for PanguWeather (who only provide a pseudo-code "implementation"), it's not for the others: FourCastNet: https://github.com/NVlabs/FourCastNet, GraphCast: https://github.com/google-deepmind/graphcast Given that GraphCast seems to outperform ClimaX, it would have been good to compare against it as well and also FourCastNet since it's one of the first papers to perform weather forecasting on such a scale.
> >
> > **Comparison with FourCastNet.** Thank you for the suggestion and for providing links to the repositories. PanguWeather has made their pre-trained model files available, but they use a different spatial resolution, so we cannot compare with them. FourCastNet similarly uses a different resolution, but we were able to retrain the FourCastNet model using our dataset for fair comparison and using their recommended hyperparameters and training.
> >
> > We report below the results that show that in general ClimODE has better RMSE accuracy (especially significant for the geopotential variable z), while the performance with respect to ACC metric is largely comparable (same or slightly worse for ClimODE). We will include this experiment in the paper based on your feedback.
> >
> > |          |                   | RMSE        |                        | ACC         |          |
> > |----------|-------------------|-------------|------------------------|-------------|----------|
> > | Variable | Lead Time (hours) | FourCastNet | ClimODE                | FourCastNet | ClimODE  |
> > | z        | 6                 | 149.4       | **102.9**   $\pm$ 9.3 | 0.73        | 0.99     |
> > |          | 12                | 217.8       | **134.8**   $\pm$ 12.3     | 0.58        | 0.99     |
> > |          | 18                | 275.0       | **162.7**   $\pm$ 14.4     | 0.58        | 0.98     |
> > |          | 24                | 333.0       | **193.4**   $\pm$ 16.3     | 0.58        | 0.98     |
> > | t        | 6                | 1.18        | **1.16**   $\pm$ 0.06      | 0.99        | 0.97     |
> > |          | 12               | 1.47        | **1.32**   $\pm$ 0.13      | 0.99        | 0.96     |
> > |          | 18               | 1.65        | **1.47**   $\pm$ 0.16      | 0.99        | 0.96     |
> > |          | 24               | 1.83        | **1.55**   $\pm$ 0.18      | 0.99        | 0.95     |
> > | t2m        | 6                | 1.28        | **1.21**   $\pm$ 0.09      | 0.99        | 0.97     |
> > |          | 12               | 1.48        | **1.45**   $\pm$ 0.10      | 0.99        | 0.96     |
> > |          | 18               | 1.61        | **1.43**   $\pm$ 0.09      | 0.99        | 0.96     |
> > |          | 24               | 1.68        | **1.40**   $\pm$ 0.09      | 0.99        | 0.96     |
> > | u10        | 6                | 1.47        | **1.41**   $\pm$ 0.07      | 0.93        | 0.91     |
> > |          | 12               | 1.89        | **1.81**   $\pm$ 0.09      | 0.91        | 0.89     |
> > |          | 18               | 2.05        | **1.97**   $\pm$ 0.11      | 0.89        | 0.88     |
> > |          | 24               | 2.33        | **2.01**   $\pm$ 0.10      | 0.89        | 0.87     |
> > | v10        | 6                | 1.54        | **1.53**   $\pm$ 0.08      | 0.95        | 0.92     |
> > |          | 12               | 1.81        | 1.81   $\pm$ 0.12      | 0.91        | 0.89     |
> > |          | 18               | 2.11        | **1.96**   $\pm$ 0.16      | 0.88        | 0.88     |
> > |          | 24               | 2.39        | **2.04**   $\pm$ 0.10      | 0.85        | 0.86     |

---

> > > ### Author Response · Authors · 2023-11-16
> > > **Response Continued**
> > >
> > > > While Figure 6 shows qualitatively the soundness of the predicted bias and variance, there's no quantitative approach that evaluates the quality of the bias and variance output by the model. A metric like CRPS (Continuous Ranked Probability Score) can showcase that.
> > >
> > > **Quantifying quality of bias and variance outputs, and CRPS plots.** Thank you for another great suggestion. It essentially boils down to whether the predicted distributions $\mathcal{N}(u_k +  \mu_k, \sigma_k^2)$. These plots indicate that the emission model is highly aligned with data, and does not indicate any immediate biases or skews. These results are averaged over space and time, and spatially local variations are still possible. The mean $\mu$ plots show that means match data well. The standard deviation $\sigma$ plots show some bimodality of predictions with either no or moderate amount of uncertainty.
> > >
> > > We also evaluated our model using CRPS (Continuous Ranked Probability Score). Our results show that ClimODE can indeed produce good estimates of variance and bias. We will include all these plots in the Appendix.
> > >
> > > CRPS Plot: https://postimg.cc/RqGc04NG
> > >
> > > Correlation Plots: https://postimg.cc/6TBmSWkh, https://postimg.cc/v47npyFW
> > >
> > > > Table 1 says that NowCastNet doesn't provide uncertainty estimation, but it's a generative model which can provide such estimates and in general approximate the true underlying distribution.
> > >
> > > **ClimODE vs. NowCastNet with respect to uncertainty estimation.**  Thank you for the opportunity to clairfy this important point. NowCastNet indeed operates as a generative model and to obtain uncertainty estimates one requires multiple samples and forward passes to obtain a scatter of outputs, whose summary statistics can be computed. However, for ClimODE, a single forward pass suffices for accurate uncertainty estimation. That is, our approach allows us to obtain reliable uncertainty estimates in a single pass without the need for repeated sampling that incurs extra costs and approximation errors.
> > >
> > > > In section 3.2, it's unclear why $\dot{\mathbf{v}}_{k}(x,t)$ $=$ $\ddot{u}_k (x,t)$  especially when $\ddot{u}_k (x,t)$ is not a vector.
> > >
> > > Good catch: this is a typo. We will remove this remark. Fortunately, this does not affect our model equations.
> > >
> > > > In section 3.6, how is $\tilde{\dot{u}}(t_0)$numerically approximated from past states?
> > >
> > > We compute $\tilde{\dot{u}}(t_0)$ by utilizing torchcubicspline (https://github.com/patrick-kidger/torchcubicspline) to fit data-points $\{u_k(t_0 -2),u_k(t_0 -1),u_k(t_0) \}$ in order to get a smooth derivative approximation.
> > >
> > > > Lacks training details for ClimaX as well as the training runtimes and number of GPUs used for ClimaX.
> > >
> > > We train ClimaX on the same set of hyperparameters as described in their official paper on our set of data splits and inputs for a fair comparison. ClimaX is trained on 4 GPUs with an approximate runtime of 24 hours.
> > >
> > > > Why not use different time resolutions for solving the ODE and assessing their effect? Same goes for the ODE-solver. Given that you state that Runge-Kutta can be used with a low computational cost, it would add more quality to the paper overall if you include it as well.
> > >
> > > We utilize one-hour time resolutions for solving the ODE system, and finer time-stepping or higher precision solvers can (and perhaps should be) used. We settled on a compromise of 1 hour increment for a second-order ODE that already provided us with good efficiency, and also empirically maintained the mass conservation (see above).
> > >
> > > > How long does it take to train?
> > >
> > > For global forecast, We train our model on a single NVIDIA Tesla V100 64GB GPU, without any parallelization techniques for 300 epochs, and it takes around 20--24 hours.
> > >
> > > We are grateful for your thoughtful comments and suggestions, which have allowed us to emphasize some salient aspects, and shed light on subtle facets of the proposed method. We hope that your concerns have been sufficiently addressed, and if so, you would consider raising your score. We would also be happy to discuss more.

---

> > > > ### Comment · Reviewer_THXC · 2023-11-17
> > > >
> > > > I thank the authors for their response and for the additional experiments. Most of my main points have been addressed, although, I would have still liked to see the CRPS for at least NowCastNet even though they require multiple forward passes to get uncertainty estimations.
> > > >
> > > > Overall, I find, with these additional experiments and clarifications that the quality of the paper will be greatly improved and thus I am updating my score to reflect that.

---

> > > > > ### Author Response · Authors · 2023-11-23
> > > > > **Response**
> > > > >
> > > > > Many thanks for your strong support. We tried conducting an experiment with NowCastNet as you suggested; but found out that the NowCastNet model specifically predicts precipitation within designated regions, and as such, it does not encompass global climate modeling. Additionally, it lacks a publicly available code implementation for training the model from the ground up.
> > > > >
> > > > > We will indeed incorporate all the additional experiments and clarifications from our response. We are indebted for your help in reinforcing the strengths of this work. Thank you very much.

---

### Official Review · Reviewer_K2uQ · 2023-10-31

**Soundness:** 3 good
**Presentation:** 4 excellent
**Contribution:** 3 good
**Rating:** 8
**Confidence:** 3

**Summary:**

This paper proposes ClimODE, a novel climate modeling approach that leverages physics-based constraints. It represents climate dynamics as a continuous-time advection process governed by partial differential equations (PDEs). The PDEs are discretized into ODEs using the method of lines, with the velocity field modeled by a neural network integrating convolutions and attention. Gaussian emission models estimate the prediction uncertainties and source variations. Empirically, ClimODE outperforms existing data-driven methods in global and regional forecasting tasks, highlighting the efficacy of continuous-time physical constraints.

**Strengths:**

1. **Physical Prior as the Foundation:**
   The paper ingeniously grounds its methodology in the continuity equation, a well-established physical prior. This not only lends an elegant formulation but also ensures an interpretable and efficient model. The fair integration of this physical prior into the deep model showcases an exemplary fusion of the first principles with modern DL techniques.

2. **Gaussian Emission Model:**
   The adoption of the Gaussian as an emission model is both impressive and reasonable. It aptly addresses uncertainties and unknown sources in climate forecasting, offering a reasonable approach to handling the inherent unpredictability of the domain.

3. **Experimentation and Insight:**
   The experimental setup and investigation presented in the work are both rigorous and enlightening. The thoroughness of the research provides valuable insights and sets an easy-to-access benchmark for future endeavors in the field.

Also, the presentation is clear and easy to follow.

**Weaknesses:**

1. For the model of FLOW VELOCITY (section 3.2), as we already know $\dot{\mathbf{v}}_k(\mathrm{x}, t) = \ddot{u}_k(\mathrm{x}, t)$,   why we still need to parameterize it? Is it because the computation of $\ddot{u}_k(\mathrm{x}, t)$ is too costly? There are also many methods that could approximate $\ddot{u}_k(\mathrm{x}, t)$ based on $\dot{u}_k(\mathrm{x}, t)$. More discussion or claims are encouraged.

2. The model treats the advection PDE as independent for each task or quantity. However, in the intricate tapestry of climate dynamics, various quantities are interdependent. For instance, wind patterns can influence temperature fluctuations.  The paper doesn't clearly illustrate how the proposed method addresses these inherent inter-correlations.

**Questions:**

See weakness

---

> ### Author Response · Authors · 2023-11-16
> **Response**
>
> Many thanks for your constructive review. We address your concerns and incorporate your suggestions below.
>
> > For the model of FLOW VELOCITY (section 3.2), as we already know $\dot{ \mathbf{v}}_{k}(x,t)$ $=$ $\ddot{u}_k (x,t)$
> why we still need to parameterize it? Is it because the computation of $\ddot{u}_k (x,t)$ is too costly? There are also many methods that could be approximated. More discussion or claims are encouraged.
>
> **Need to parameterize the flow velocity.** Thank you for raising an excellent point. In an advection system the climate quantities $u(\mathbf{x},t)$ are transported (moved and redistributed) around by a flow velocity $\mathbf{v}(x,t)$, which is a vector field that points where the climate `mass' is moving. Given a known starting state $u$, the system evolution is fully determined by the velocity $\mathbf{v}$ by solving eq (2) forward in time.
>
> The velocity field function is generally unknown, and we thus learn it from data by parameterising the the velocity function as a neural network $\mathbf{v}(\mathbf{x},t | \theta)$. We also note that the remark that $\dot{\mathbf{v}}_k = \ddot{u}_k$ was a typo, and in general we cannot solve for $\mathbf{v}$ by, e.g., interpolating the value $u$.
>
> Apologies for any confusion: we have fixed this typo and our model equations remain unaffected.
>
> > The model treats the advection PDE as independent for each task or quantity. However, in the intricate tapestry of climate dynamics, various quantities are interdependent. For instance, wind patterns can influence temperature fluctuations. The paper doesn't clearly illustrate how the proposed method addresses these inherent inter-correlations.
>
> **Explicating how ClimODE accounts for variable correlations.** This is another great point that we elucidate here. We present our equations for a single quantity $k$ at a time (for instance, equation (2)) because the convenient vector calculus notation $\dot{u}_k - \nabla \cdot (u_k \mathbf{v}_k)$ applies only to scalar $u$, and a `batch' version $\dot{\mathbf{u}}$ does not admit elegant notation. Nevertheless, our model is still able to consider the entire state vector $\mathbf{u} = (u_1, \ldots, u_K)$ at the same time, and all $K=5$ climate quantities are affected by each other. This can be seen in equation (4), where the velocity change $\dot{\mathbf{v}}_k = f( \mathbf{u}, \ldots)$ of quantity $k$ depends on all state variables, and thus subsequently $\dot{u}_k$ also depends on all state variables. Thus, for instance, wind can affect temperature and vice versa.
>
> To demonstrate the emerging couplings of quantities (ie. wind, temperature, pressure potential), we plot below the emission model  $\mathbf{u}^{\mathrm{pred}}(\mathbf{x},t) \in \mathbb{R}^5$ pairwise densities averaged over space $\mathbf{x}$ and time $t$. These effectively capture the correlations between quantities in the simulated weather states. For example, these densities reveal that temperatures (t,t2m) and potential (z) are highly correlated and bimodal; the horizontal and vertical wind directions are independent (u10,v10); and there is little dependency between the two groups. We will include these plots in the appendix.
>
> Plot: https://postimg.cc/0McVK5r5
>
> Again, thank you so much for your extremely constructive feedback. We hope we have sufficiently addressed your concerns, and would appreciate your stronger support for this paper.

---

> > ### Comment · Reviewer_K2uQ · 2023-11-20
> >
> > Thanks for the authors' retailed response. It resolved my concerns and I still strongly support this great work.

---

> > > ### Author Response · Authors · 2023-11-23
> > > **Response**
> > >
> > > We are glad to note that all your concerns are resolved. We are grateful for your strong support for this work. Thank you so much.

---

### Official Review · Reviewer_F1k5 · 2023-10-31

**Soundness:** 4 excellent
**Presentation:** 4 excellent
**Contribution:** 3 good
**Rating:** 8
**Confidence:** 3

**Summary:**

This paper introduces ClimODE, a neural ODE combining a convolutional local mechanism and a global attention mechanism to predict one timestep of weather evolution. It uses ERA5 data from WeatherBench 1 and compares it to ClimaX model and against a standard Neural ODE.

**Strengths:**

Originality
- First NeuralODE work applied to this problem
- Extending Neural ODEs to be more effective for the specific climate problem

Quality
- High-quality writing, figures
- Using relevant, real-world data for evaluation

Clarity
- Written well and understandable
- Detailed explanations

Significance
- Satisfies the need for ML climate models to be value-conserving and have a probabilistic forecast
- Contributes towards faster climate and weather modeling, potentially using less computational resources

**Weaknesses:**

- A couple of Figures (e.g Figure 5) would benefit from a longer caption/description
- Not comparing against state-of-the-art methods, such as GraphCast
- It should be compared against pre-trained ClimaX or other methods that don't require pre-training

**Questions:**

1. What do you mean by one-shot GAN referring to Ravuri et al., 2021? Is there any pre-training involved
2. There is Weatherbench 2 available now, maybe to recent to be included in this submission, but should be mentioned in future work: https://arxiv.org/abs/2308.15560, https://sites.research.google/weatherbench/
3. Are any of the competing methods such as GraphCast, PanguWeather, FourCastNet that now available?
4. Is the comparison against ClimaX fair? The main idea of ClimaX is to use pre-training, but you state you use all methods without pertaining
5. The statement "IFS is still far ahead of any deep learning method" doesn't really hold anymore, e.g.: https://arxiv.org/pdf/2307.10128.pdf
6. Kreislers GNN work should be mentioned here as well: https://arxiv.org/pdf/2202.07575.pdf
7. This work could be useful too: https://arxiv.org/pdf/2304.04664.pdf

---

> ### Author Response · Authors · 2023-11-16
> **Response**
>
> Many thanks for your thoughtful feedback and suggestions. We address all your comments below.
>
> ##  Weaknesses:
>
> > A couple of Figures (e.g Figure 5) would benefit from a longer caption/description
>
> Thanks for the feedback. We will add more details to the captions of fig 3 (the pipeline), fig 5 (ODE ablations) and fig 6 (emissions over time) to make them more descriptive. For instance, fig 5 caption will be expanded to
>
> **The effect of including different ClimODE components on RMSE**. An ablation showing how enhancing the vanilla neural ODE (blue) incrementally with advection (orange), global attention (green) and emission (red) leads to progressively better RMSE scores. We note that that advection yields most improvement in accuracy, while attention turns out to be the least important.
>
>
> ##  Questions:
>
> > What do you mean by one-shot GAN referring to Ravuri et al., 2021? Is there any pre-training involved
>
> Ravuri et al., 2021 propose a GAN framework to model weather prediction as a density estimation problem, where the network learns to predict the weather state at a fixed increment in time. One-shot emphasizes that their model does not unroll the weather state forward in multiple short time steps, instead taking a single leap. They do not perform pre-training and demonstrate their method by modeling precipitation over local regions.
>
> > There is Weatherbench 2 available now, maybe to recent to be included in this submission, but should be mentioned in future work: https://arxiv.org/abs/2308.15560, https://sites.research.google/weatherbench/
>
> Many thanks for the reference. We will include a citation and position appropriately in the paper.
>
>
> > Is the comparison against ClimaX fair? The main idea of ClimaX is to use pre-training, but you state you use all methods without pertaining
>
> Thank you for the opportunity to clarify this. We trained the ClimaX model from scratch using the same dataset as for ClimODE without pre-training to compare how much signal the two methods are able to extract from the same data. Comparing to pre-trained ClimaX would mean comparing to a model that has seen much more data than us, which would give unfair advantage to ClimaX. We acknowledge that pretraining can be effective for a practical weather predictor, and are very keen to leverage pre-training in future with larger scale implementations of our advection ODE model.
>
>
> > The statement "IFS is still far ahead of any deep learning method" doesn't really hold anymore, e.g.: https://arxiv.org/pdf/2307.10128.pdf, Kreislers GNN work should be mentioned here as well:https://arxiv.org/pdf/2202.07575.pdf
>
> Thank you very much for these  references. We will add these and update our statements about IFS.

---

> > ### Author Response · Authors · 2023-11-16
> > **Response Continued**
> >
> > > Are any of the competing methods such as GraphCast, PanguWeather, FourCastNet that now available?
> >
> > PanguWeather have not published their code, but released a pre-trained model file that can be used to forecast. However, we cannot evaluate our data with PanguWeather since, differently from us, their model file supports only a very high spatial resolution Earth grid of size (720, 1441).
> >
> > GraphCast have made a public implementation, but we found it to be incomplete and so were not able to make it work despite our efforts.
> >
> > **Comparison with FourCastNet.** FourCastNet also trains with a higher spatial resolution, so we cannot use their pretrained models either. Fortunately, FourCastNet has released their implementation and recommended setting of hyperparameters. Based on your feedback, we produced a new experiment where we trained the FourCastNet model according to their training scheme on our dataset.
> >
> > We report these new results below. We note that ClimODE compares favorably with FourCastNet in terms of RMSE: in particular, our model achieves significantly better RMSE for the $z$ (i.e., geopotential) variable. The ACC results are largely competitive across variables for the two methods.   We will include these results in the paper.
> >
> >
> > |              |                   | RMSE        |                        | ACC         |          |
> > |----------|-------------------|-------------|------------------------|-------------|----------|
> > | Variable | Lead Time (hours) | FourCastNet | ClimODE                | FourCastNet | ClimODE  |
> > | z        | 6                 | 149.4       | **102.9**   $\pm$ 9.3 | 0.73        | 0.99     |
> > |          | 12                | 217.8       | **134.8**   $\pm$ 12.3     | 0.58        | 0.99     |
> > |          | 18                | 275.0       | **162.7**   $\pm$ 14.4     | 0.58        | 0.98     |
> > |          | 24                | 333.0       | **193.4**   $\pm$ 16.3     | 0.58        | 0.98     |
> > | t        | 6                | 1.18        | **1.16**   $\pm$ 0.06      | 0.99        | 0.97     |
> > |          | 12               | 1.47        | **1.32**   $\pm$ 0.13      | 0.99        | 0.96     |
> > |          | 18               | 1.65        | **1.47**   $\pm$ 0.16      | 0.99        | 0.96     |
> > |          | 24               | 1.83        | **1.55**   $\pm$ 0.18      | 0.99        | 0.95     |
> > | t2m        | 6                | 1.28        | **1.21**   $\pm$ 0.09      | 0.99        | 0.97     |
> > |          | 12               | 1.48        | **1.45**   $\pm$ 0.10      | 0.99        | 0.96     |
> > |          | 18               | 1.61        | **1.43**   $\pm$ 0.09      | 0.99        | 0.96     |
> > |          | 24               | 1.68        | **1.40**   $\pm$ 0.09      | 0.99        | 0.96     |
> > | u10        | 6                | 1.47        | **1.41**   $\pm$ 0.07      | 0.93        | 0.91     |
> > |          | 12               | 1.89        | **1.81**   $\pm$ 0.09      | 0.91        | 0.89     |
> > |          | 18               | 2.05        | **1.97**   $\pm$ 0.11      | 0.89        | 0.88     |
> > |          | 24               | 2.33        | **2.01**   $\pm$ 0.10      | 0.89        | 0.87     |
> > | v10        | 6                | 1.54        | **1.53**   $\pm$ 0.08      | 0.95        | 0.92     |
> > |          | 12               | 1.81        | 1.81   $\pm$ 0.12      | 0.91        | 0.89     |
> > |          | 18               | 2.11        | **1.96**   $\pm$ 0.16      | 0.88        | 0.88     |
> > |          | 24               | 2.39        | **2.04**   $\pm$ 0.10      | 0.85        | 0.86     |
> >
> > We are grateful for your constructive review. We hope that our response, including the new experimental results comparing ClimODE with FourCastNet, addresses your concerns and reinforces your support for this work.

---

> > > ### Author Response · Authors · 2023-11-21
> > > **Gentle Reminder**
> > >
> > > Many thanks for you insightful suggestions and review. We believe that we have clarified the points being raised and addressed of all of your concerns. If there are any further points, let us know as we are fully committed to discuss more.

---

### Author Response · Authors · 2023-11-22
**General Response**

We are grateful to the reviewers for their insightful comments and constructive suggestions and to the (senior) area, program, and general chairs for their great service to the community.

Here we summarize how the review process has helped consolidate and reinforce the key contributions of this work.

## Positive aspects

We are glad that the reviewers unanimously liked the paper and have recognized numerous strengths of this work. They particularly appreciated its clear contribution as a pioneering work in extending Neural ODEs to model climate. The novelty of incorporating physical priors in climate modeling and aspects such as uncertainty quantification were also highlighted. Additionally, the paper was commended for its strong empirical performance, efficient parameterization, rigorous experimentation, and insightful analysis.

## Questions and inputs for further improvement.

The reviewers brought up several valuable points, focusing in particular on comparisons with large-scale models and the presentation of results for longer lead time predictions. In response, we have meticulously acted on their feedback, outlining the additional analyses and experiments we conducted to address these issues. All these supplementary analyses and experiments will be thoroughly incorporated in the final version of the paper.

**Comparison with large-scale model**

In response to the reviewers' recommendation for a comparison with a large-scale NWP model, we conducted a comparative analysis with FourCastNet. Following their suggestion, we re-trained the FourCastNet model on our dataset, utilizing the hyperparameters provided in their official paper. It is noteworthy that ClimODE exhibits favorable performance compared to FourCastNet, particularly in terms of Root Mean Square Error (RMSE) for the $z$ variable, representing geopotential. Our model achieves significantly improved RMSE for this variable. Additionally, both models' Accuracy (ACC) results demonstrate competitive performance across various variables.

**Longer lead-time predictions**

Acting on the reviewers' another excellent suggestion, we carried out additional experiments focusing on longer lead-time predictions, specifically for 6-day and 3-day lead times. In this extended analysis, we included ClimaX predictions for comparative purposes. The findings indicate that temperature and potential variables (t, t2m, z) exhibit a relatively stable performance over longer forecast periods. However, it is observed that the reliability of wind direction variables (u10, v10) diminishes over extended time horizons. It's worth noting that ClimaX, while demonstrating remarkable stability in long-term predictions, exhibits lower performance than ClimODE. This suggests that ClimODE outperforms ClimaX, particularly in scenarios involving longer lead-time predictions.

We also address all specific questions, comments, and concerns raised by reviewers in our detailed individual responses for each reviewer.

Climate prediction is arguably one of the most important challenges and opportunities for machine learning today and requires a collective effort and active engagement from our community. We're particularly pleased to acknowledge all the reviewers again for the very high quality of reviews across the board that have helped us take a step forward in this pursuit. Thank you very much!

---

### Meta-Review · Area_Chair_RF59 · 2023-12-05

**Metareview:**

This paper introduces a weather and climate modeling approach that combines a spatiotemporal continuous-time process for modeling advection, value-conserving dynamics and uncertainty in predictions via a Gaussian emission model. The model is validated on ERA5 data from WeatherBench.

Strengths:
* Successful application of NeuralODEs to climate
* Strong physical priors
* Comparison to ClimaX and to FourCastNet (during rebuttal period) which it outperforms
* Clarity and execution of the paper

Weaknesses:
* Lack of comparison to GraphCast, PanguWeather or some pretrained foundation models like ClimaX, and forecasts at a coarser scale than SOTA

The authors clarified many points during the rebuttals and discussion.

Based on these unanimous scores (8), the paper deserves publication as an oral.

**Justification For Why Not Higher Score:**

N/A

**Justification For Why Not Lower Score:**

The work is of very high scientific value and execution, praised by unanimous reviewers.

---

### Decision · Program_Chairs · 2024-01-16

Accept (oral)